# Multi-Robot Coverage Path Planning for the Inspection of Offshore Wind Farms: A Review

Ashley J. I. Foster [1], Mario Gianni [2], Amir Aly [1], Hooman Samani [3] and Sanjay Sharma [1,*]

1 School of Engineering, Computing and Mathematics, Faculty of Science and Engineering, University of Plymouth, Plymouth PL4 8AA, UK; ashley.foster@postgrad.plymouth.ac.uk (A.J.I.F.); amir.aly@plymouth.ac.uk (A.A.)
2 School of Electrical Engineering, Electronics and Computer Science, University of Liverpool, Liverpool L69 3BX, UK; mario.gianni@liverpool.ac.uk
3 Creative Computing Institute, University of the Arts London, London SE5 8UF, UK; h.samani@arts.ac.uk
* Correspondence: sanjay.sharma@plymouth.ac.uk

**Abstract:** Offshore wind turbine (OWT) inspection research is receiving increasing interest as the sector grows worldwide. Wind farms are far from emergency services and experience extreme weather and winds. This hazardous environment lends itself to unmanned approaches, reducing human exposure to risk. Increasing automation in inspections can reduce human effort and financial costs. Despite the benefits, research on automating inspection is sparse. This work proposes that OWT inspection can be described as a multi-robot coverage path planning problem. Reviews of multi-robot coverage exist, but to the best of our knowledge, none captures the domain-specific aspects of an OWT inspection. In this paper, we present a review on the current state of the art of multi-robot coverage to identify gaps in research relating to coverage for OWT inspection. To perform a qualitative study, the PICo (population, intervention, and context) framework was used. The retrieved works are analysed according to three aspects of coverage approaches: environmental modelling, decision making, and coordination. Based on the reviewed studies and the conducted analysis, candidate approaches are proposed for the structural coverage of an OWT. Future research should involve the adaptation of voxel-based ray-tracing pose generation to UAVs and exploration, applying semantic labels to tasks to facilitate heterogeneous coverage and semantic online task decomposition to identify the coverage target during the run time.

**Keywords:** multi-robot; coverage path planning; UAV; structural inspection; offshore wind



## 1. Introduction

Offshore wind turbine inspection is an area of increasing interest with the increasing prevalence of wind power [1]. The relevance of renewable offshore energy sources has never been greater than at present [2]. Offshore wind has several benefits when compared to onshore wind turbines [3]. Offshore wind farms experience greater and more predictable wind speeds with reduced turbulence, ensuring that a single OWT is more productive than an onshore counterpart. Additionally, offshore farms do not need to compete with other land uses and are less likely to meet resistance from local communities. While there are significant benefits to offshore renewable wind energy, so, too, are there serious challenges to overcome. Dynamic loads from wind and waves, as well as saltwater, damage and degrade offshore turbines quicker than their onshore counterparts. Installation is significantly more expensive than on shore, and operation and maintenance (O&M) operations are considerably more complicated. Within the already growing offshore wind sector, O&M is predicted to become the second largest subsector of the offshore renewable market in the UK by 2030 and potentially the largest in the 2040s [4]. O&M can be broken down into four subsections [4], details of which are shown in Figure 1.

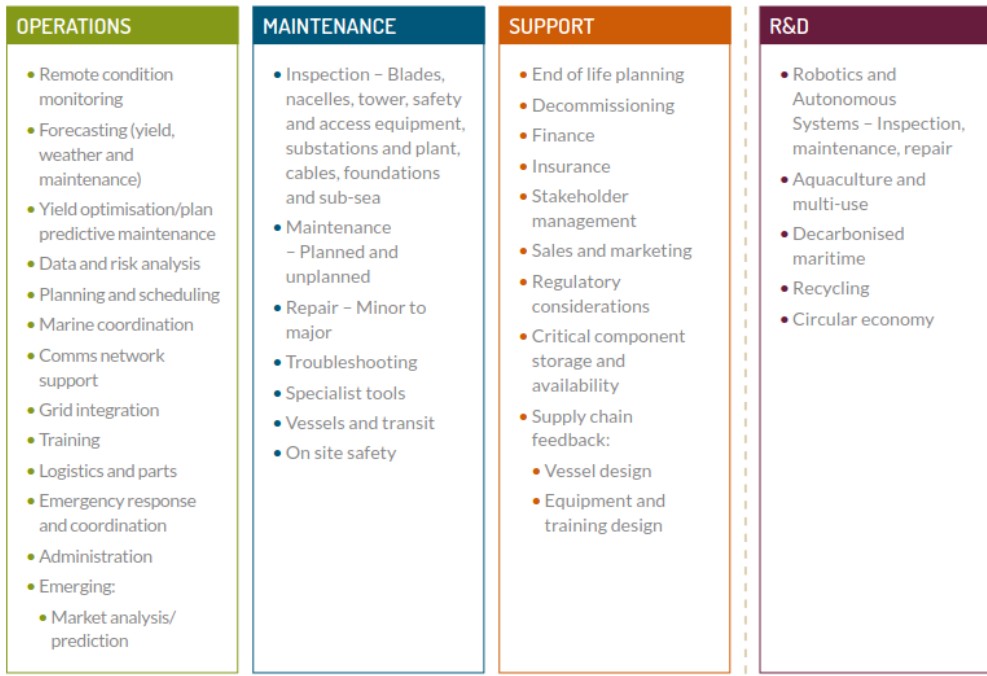

**Figure 1.** Key services that make up offshore wind O&M [4].

Maintenance is a particularly high-risk aspect of O&M, involving highly skilled technicians out in the field for long periods, undertaking maintenance work on the turbines. These services can include rappelling to inspect or repair blades and diving to inspect cabling, all the while being far away from any emergency assistance. The teams undertaking these operations are composed of disparate highly qualified technicians. Using traditional methods, a turbine inspection with three technicians takes 3–6 h, allowing time for only two to three turbines to be inspected in a day [5]. Considering that wind farms can often house hundreds of turbines, the cumulative labour time required for a single wind farm's regular inspections can commonly reach thousands of hours. It is both this financial cost and the human risk that incentivise the use of robots. Commercial remotely operated systems are now reasonably commonplace for offshore inspections. Remotely operated underwater vehicle (ROV) services facilitate inspections of the anchors, as well as subsea cabling [6–10]. Several companies offer unmanned aerial vehicle (UAV) services for visual and thermal imaging inspections [11–13], and recently, climbing robots have been made available for the cleaning [14] and resurfacing of OWTs [15]. Nordin et al. identified that individual unmanned vehicles have limited capacity to perform unmanned O&M for offshore wind turbines [16]; instead, the task lends itself to multi-robot systems. Five motivations for developing multi-robot systems were identified by Parker et al. [17]: (1) The task complexity is too high for a single robot; (2) the task is inherently distributed; (3) the use of several less powerful robots is often less resource-intensive than using a single powerful robot; (4) multiple robots can solve problems faster using parallelism; and (5) using multiple robots increases robustness through redundancy. Notably, each of these could apply to offshore wind O&M. Indeed, multi-robot approaches to wind energy O&M have been researched, albeit overlooking critical factors such as communication challenges and harsh environmental conditions necessary for real-world implementation [18–20]. Approaches to multi-robot navigation in extreme environments require mechanisms to minimise interference and spatial conflicts [21], or else the system may perform unreliably. One interesting commonality in the aforementioned research is the use of robot heterogeneity. Parker defines robot heterogeneity as variety in terms of robot behaviour, morphology, performance quality, size, and cognition [17] within a team. Certain inspections may require a heterogeneous team, while others may be performed faster with robots specialised for certain tasks. Considering a comprehensive inspection of an OWT (one covering the

turbine's surface, the foundations, local cabling, and the turbine interior), a range of robots with varying morphologies and locomotion and sensing capabilities would be required. Variety in terms of performance quality can also affect the quality of the inspection; for example, a UAV fitted with a high-resolution camera would be able to capture footage of the same quality as one with a lower-resolution camera at a greater rate. Another possibility within a heterogeneous team is having robots collaborate in such a way as to complete tasks impossible for just one robot. Jiang et al. provided an example of just this, with a UAV being used to transport a BladeBUG to and from a wind turbine blade [22]. The authors made use of GNSS to position the UAV near the landing target, then made use of LiDAR data to position the vehicle for landing and deployment of the BladeBUG on the blade. Reaching the blade with the BladeBUG would have been impossible alone but was made possible via the UAV. Another example of such behaviour is using an unmanned surface vehicle (USV) as a mother ship for UAVs, with the USV serving as a "marsupial" robot. Fan et al. were concerned with the autonomous landing of a UAV on a USV using a fuzzy self-adaptive PID controller [23]. A marsupial relationship was also detailed by Miškovic et al., who reported a USV that relies on a UAV to localise itself with respect to a floating object needing tugging [24]. Zhang et al. described a fully autonomous system for the recovery of fixed-winged UAVs, making use of an arresting cable and a net to safely land the UAVs [25].

The use of multi-robot teams for offshore inspection is currently a sparse area of research. In this work, the focus is on visual inspection of OWTs. It is common for the operators of OWTs to request an inspection at the end of the warranty and regularly every three years thereafter [26]. A typical inspection requires the capture of high-quality images of each side of the OWT blade (suction side, pressure side, leading edge, and trailing edge). Tower and nacelle inspections are sometimes also required and are concerned with identifying welding defects, coating issues, and mechanical damage [27]. There may be areas of particular interest, such as the blades, although this can be seen as a variation of the problem. Further inspection operations use USVs or ROVs to inspect the floating substructures of floating OWTs or the underwater cabling [28]. The ORE Catapult Levenmouth demonstration turbine detailed in Figure 2, built to facilitate OWT research, provides an example of the structure to be inspected. These operations all involve capturing images of all of an area of interest; the problem of ensuring the entirety of an area of interest is covered by a sensor's footprint is known as the coverage path-planning problem.

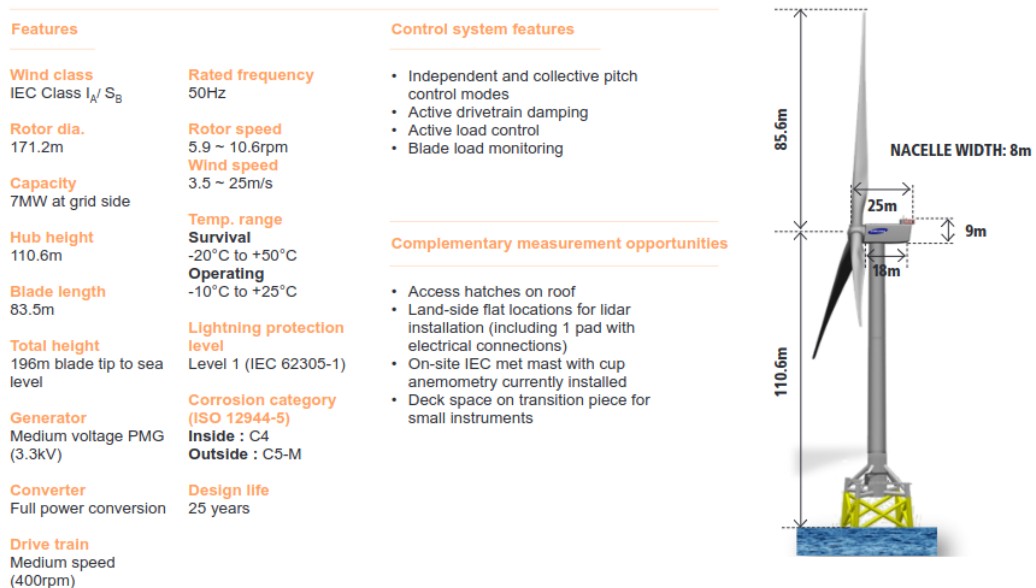

**Figure 2.** Specifications of the Levenmouth 7 MW demonstration OWT [29].

Coverage path planning, as defined by Choset [30], is the problem of passing over all points in the target environment. While at the time, coverage was mostly concerned with the coverage of 2D planes, such as mowing a lawn [31] or vacuuming a floor [32], the definition of coverage has now expanded to include 3D environments and structural coverage. To this end, Almadhoun et al. define coverage path planning as "a process of exploring or exhaustively searching a workspace, whether it a structure of interest or an environment and determining in the process the set of locations to visit while avoiding all possible obstacles" [33]. While reviews of the literature surrounding coverage more generally exist, such as Choset's inaugural survey of robotics for coverage [30] and Almadhoun et al.'s survey on multi-robot coverage path planning for model reconstruction and mapping [33], these surveys did not focus on the domains representing offshore wind inspection, a specific variation of the problem characterised by its environmental lack of structure and sparseness. To the best of our knowledge, this work provides the first literature review of coverage path planning for OWT inspections.

This paper is structured as follows: Section 2 details the methodology used to conduct the literature review; Section 3 covers the approach undertaken for analysis of the works retrieved from the literature search; Section 4 provides an analysis of approaches to environmental modelling used in the retrieved works; Section 5 covers the approaches to decision making and their applicability to offshore O&M; Section 6 is concerned with coordination approaches used in the literature and their applicability; finally, the work is concluded, and future research directions are discussed. The contributions of this work include the first systematic literature review of multi-robot coverage, following a strict systematic procedure novel to robotics. A taxonomy and discussion of current works are provided, and several gaps in current research and avenues for the future are identified.

## 2. Methodology

A review was conducted to identify the research gaps in the literature on multi-robot coverage for offshore wind inspection [34]. To ensure the quality of this review, the PRISMA 2018 checklist for scoping reviews was followed [35]. The review was structured according to the PICo framework for qualitative reviews, as detailed in Table 1.

**Table 1.** PICo definitions for environmental representations with search concepts.

| P | I | Co |
|:---:|:---:|:---:|
| **Population** | **Interest** | **Context** |
| Multi-robot systems | Coverage | Unknown and unstructured environments |
| **Search Concepts** | | |
| Multi-robot | Coverage | Unknown and unstructured |
| **Alternative Terms** | | |
| Multi-agent | | Unknown Unstructured Extreme Real |

Based on this framework, the research question and subquestions were formulated after a brief review of the literature as shown in Table 2.

An advanced search was conducted in the IEEE Xplore, The ACM Guide to Computing Literature, Scopus, and Web of Science databases. Details of these libraries are presented in Table 3. These four databases provide time-efficient access to a wide range of peer-reviewed publications.

**Table 2.** Research questions.

| Research Question |
| --- |
| What is the most suitable framework for multi-robot coverage in domain applications resembling offshore wind inspection? <br> Subquestions: <br> • What is the most suitable environmental model for multi-robot coverage in terms of suitability for domain applications resembling offshore wind inspection? <br> • What is the most suitable multi-robot coverage decision making approach for domain applications resembling offshore wind inspection? <br> • What is the most suitable strategy to effectively coordinate a multi-robot system for domain applications resembling offshore wind inspection? |

**Table 3.** Digital libraries used in the review.

| Digital Library | Description | URL | Area of Focus |
| --- | --- | --- | --- |
| IEEE Xplore | A digital library provides all IEEE publications, as well as those from its publishing partners. | https://ieeexplore.ieee.org/ (accessed on 20 September 2023) | Computer science, electrical engineering, and electronics. |
| The ACM Guide to Computing Literature | The Association of Computing Machinery's digital library provides all ACM publications and works from all major publishers. | https://dl.acm.org/ (accessed on 20 September 2023) | Computing and Information Technology |
| Scopus | Scopus covers 240 disciplines to ensure researchers, instructors, librarians, and students have confidence that they are not missing out on the vital information they need to advance their research and scholarship. | https://www.scopus.com/ (accessed on 20 September 2023) | General |
| Web of Science | The Web of Science is a paid-access platform that provides access to multiple databases that provide reference and citation data from academic journals, conference proceedings, and other documents in various academic disciplines. | https://www.webofscience.com/wos/ (accessed on 20 September 2023) | General |

Based on the research question, a query was formed, as shown in Table 4. Due to the nature of offshore wind inspection, a search query making use of the term "offshore" would have yielded no results due to the lack of current research.

**Table 4.** Search query.

| Search Query |
| --- |
| • (multi-robot* OR multi-agent*) AND (coverage) AND ((unstructured AND environment*) OR (unknown AND environment*) OR (extreme AND environment*) OR (real AND environment*)) |

It is the aim of this paper to identify works relevant to the OWT coverage problem and to synthesise the knowledge from these works through the lens of the OWT coverage problem. To identify works relevant—albeit not specific—to OWT coverage, works with "domain applications resembling offshore wind inspection" were identified. In examining which domain applications closely mirror offshore wind inspection, it is crucial to understand the unique characteristics of the offshore wind inspection environment. Offshore wind farms are vast and dispersed and composed of repeated turbines, usually at regular intervals but not always so. The environment can be considered sparse and unstructured in that regard. Given the nature of the energy being captured by the turbines, these areas are also highly exposed and prone to unpredictable weather; hence, we can consider the environment extreme. To ensure that the reviewed works represented the state of the art, only works published after 2015 were considered, which was achieved through filtering in the individual databases.

The exclusion criteria presented in Table 5 were formed to remove works that were not relevant despite not being excluded in screening.

**Table 5.** Criteria for article exclusion.

| Criteria Type | Included | Excluded |
| --- | --- | --- |
| Coverage control | Works considering the coverage path-planning problem | Works considering the coverage control problem |
| Environmental structure | Works considering environments resembling OTW inspection, namely unstructured, unknown, extreme, or real environments | Works considering environments not fulfilling these criteria |
| Surveys | Any non-survey work | Surveys |

The PRISMA flow chart in Figure 3 shows the number of records identified by the search strategy for each database. Initially, works duplicated across the searches were removed. The screening process was carried out by removing works whose title or abstract made no mention of "Coverage", "Multi-robot", or "Multi-agent". The works were then sought for retrieval and discarded if unavailable or requiring purchase. Finally, the literature exclusion criteria were used to remove irrelevant works.

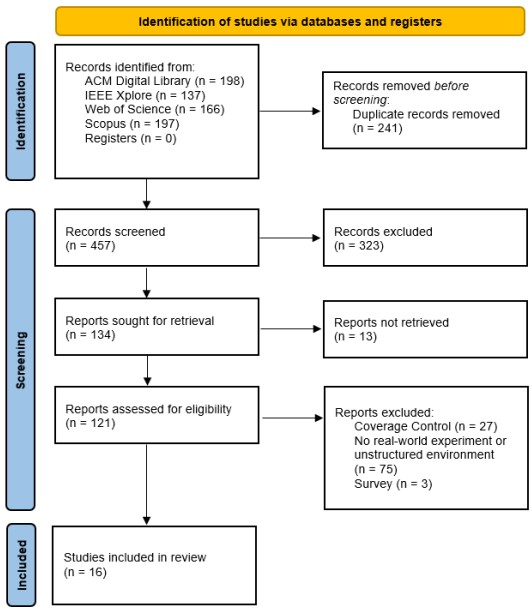

**Figure 3.** PRISMA flowchart showing the exclusion process.

## 3. Analysis

The 16 studies included in the review were then analysed. In this section, the process of analysis of these studies is detailed. As detailed in the research subquestions in Table 2, three aspects of the coverage problem are considered: (1) environmental modelling, (2) decision making, and (3) coordination.

Environmental modelling is concerned with the methods used by robots or a central planner to represent the environment and tasks within it. A taxonomy was constructed to systematically categorise and analyse the approaches featured in the studies as shown in Figure 4. This taxonomy of environmental models includes the following categories: geometric maps, featured in four studies; topological maps, featured in three studies; grid maps, featured in six studies; voxel-based maps, featured in three studies; occupancy grids, featured in three studies; and cost maps, featured in one study.

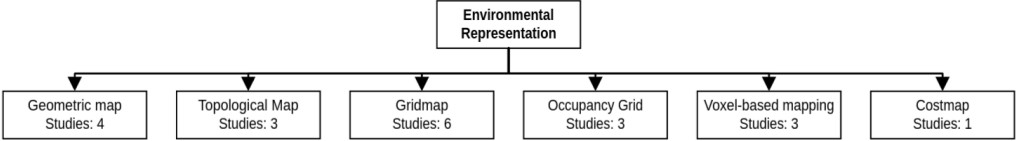

**Figure 4.** Environmental representation taxonomy with instances of reviewed surveys.

Some approaches involve the use of multiple methods and therefore appear twice. Using the taxonomy and the details of the reviewed studies, the approaches judged most suitable for OWT inspections were identified and discussed.

Decision making, as per our definition, corresponds to the collective choices defined by specified objectives made by a multi-robot system. The applicable studies were analysed using the model/non-model distinction proposed by Almadhoun et al. [33] and planning definitions posited by Yan et al. [36]. Almadhoun et al. identified a classification of approaches based on their assumed prior knowledge. Model-based approaches involve prior knowledge of the tasks and environmental structure before the coverage task. Non-model-based approaches forgo this assumption and require modelling of the environment during the task. Yan et al. identified three components that compose mobile multi-robot task-planning approaches: task decomposition, task assignment, and motion planning. Figure 5 details the analysed aspects of Decision Making. Task decomposition is not always necessary, depending on the prior knowledge, but refers to the decomposition of a multi-robot task into a set of single-robot tasks. In the case of coverage, task decomposition takes the form of decomposing the environmental representation into robot positions or poses as tasks. A task decomposition taxonomy was formed to analyse the approaches' suitability for OWT coverage as shown in Figure 6.

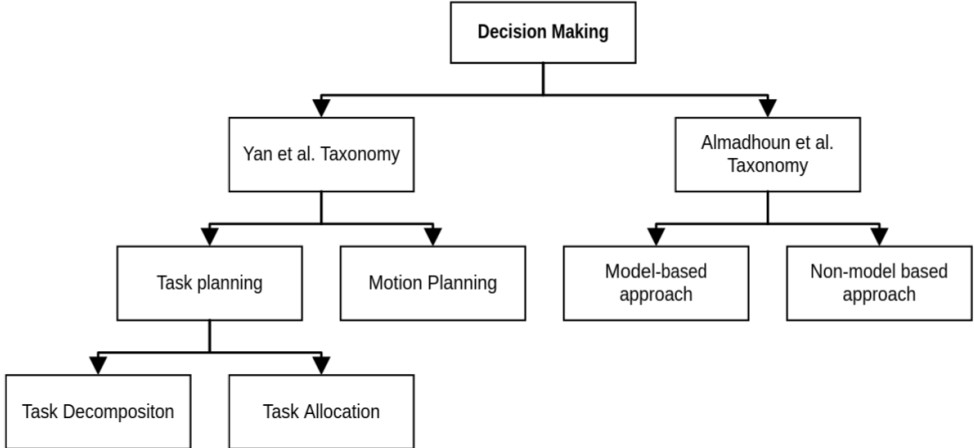

**Figure 5.** Analysed aspects of decision making including the Yan et al. taxonomy [36] and the Almadoun et al. taxonomy [33].

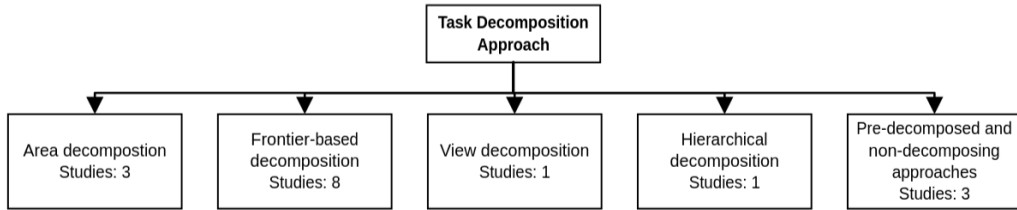

**Figure 6.** Task decomposition taxonomy with instances of reviewed surveys.

Task allocation assumes a set of single-robot tasks and is concerned with how the set of tasks can be optimally assigned to the robots. Finally, motion planning is how, given the task assignments per robot, a path for the robots can be constructed to complete all tasks optimally. Motion planning could be considered a single-robot problem regarding the order of completing the tasks but, at a lower level, necessitates collision avoidance between team members.

Farinelli et al. [37] consider coordination to be a form of cooperation, where team members consider other team members in their actions to increase system performance. Yan et al. [36] defined coordination as planning to deal with resource conflicts among team members. The aspects considered coordination in this work are aspects related to potential resource and reliability issues that may arise from the use of real robots, communication approaches, team hierarchies, fault tolerance, and robot heterogeneity. Data relating to these aspects of the studies were extracted from the works where approaches were specified. The aspects of Coordination are visualised in Figure 7.

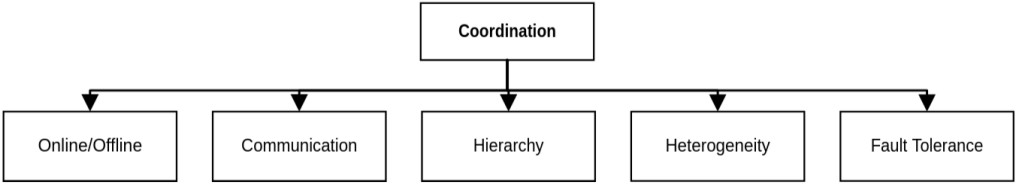

**Figure 7.** Analysed aspects of coordination.

## 4. Environmental Models

This section aims to answer the following subquestion: *What is the most suitable environmental model for multi-robot coverage in terms of suitability for domain applications resembling offshore wind inspection?* Wind farms are sparse, unstructured, and dynamic environments. There are both the predictable dynamics of the rotation of the blades and unpredictability in the current yaw orientation of the hubs. The turbines are usually spread over a kilometre away from one another, resulting in large, sparse areas in an environmental model. There may be a degree of uncertainty in GPS localisation due to the multi-path error resulting from signals reflecting off the turbines and the sea itself [38]. Therefore, in approaching this sub-question, we should view the applicability of the models through the lens of multi-robot offshore wind inspection. To analyse the approaches used in the literature and best identify those models most suited to domain applications resembling offshore wind inspection, a taxonomy of models was constructed, as seen in Table 6. These classes of approaches were then described concerning the specific implementations, followed by a discussion of the applicability of the reviewed approaches to domain applications resembling offshore wind inspection. Burgard et al. [39] identified three main challenges in constructing or choosing environmental models: (1) such models should be compact; (2) they should be task/application-dependent; and (3) given that they are constructed from sensor data, they should account for the uncertainty inherent in sensors and state estimation. An appropriate model for the offshore inspection task should consider these three factors.

**Table 6.** Environmental models used in the reviewed works.

| Environmental Model | Work (Authors, Year) |
| --- | --- |
| Geometric map | Ball et al., 2015 [40]<br>Masehian et al., 2017 [41]<br>Karapetyan et al., 2018 [42]<br>Tang et al., 2022 [43] |
| Topological map | Ball et al., 2015 [40]<br>Karapetyan et al., 2018 [42]<br>Kim et al., 2022 [44] |
| Grid map | Kalde et al., 2015 [45]<br>Song et al., 2015 [46]<br>Perez-Imaz et al., 2016 [47]<br>Sharma et al., 2016 [48]<br>Zhang et al., 2019 [49]<br>Yu et al., 2023 [50] |
| 2D cost map | Ball et al., 2015 [40] |
| Occupancy grid | Colares and Chaimowicz 2016 [51]<br>Bramblett et al., 2022 [52]<br>Kim et al., 2022 [44] |
| Octomap | Dornhege et al. 2016 [53]<br>Dong et al., 2019 [54] |
| Euclidean signed distance field (ESDF) map | Bartolomei et al., 2023 [55] |

### 4.1. Geometric Map

In some approaches, usually where the environment is known a priori, a geometric map is used. In such approaches, the environment's shape and obstacles within the environment are modelled as polygons, an example of which can be seen in Figure 8. In both [40,42], the geometric map was known a priori, representing the environment to be covered, and in both approaches, the authors used boustrophedon cell division to discretise the area into cells in a topological graph. Another method of discretisation involves overlaying a grid on the model to form a grid map and a rasterisation process, which was used by Tang et al. [43].

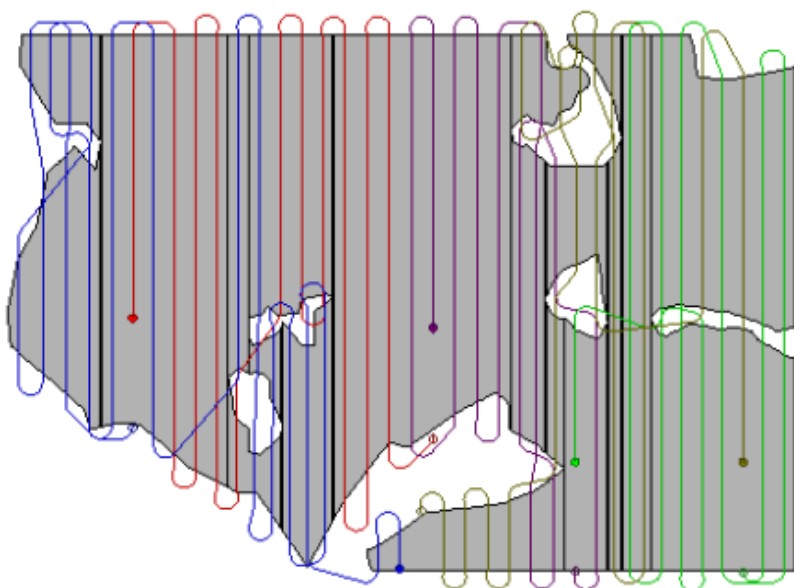

**Figure 8.** Geometric map with robot paths [42].

### 4.2. Topological Map

Choset et al. [56] defined topological representations to represent environments with graph-like structures, with the nodes representing "something distinct" and the edges representing the spatial relationship between nodes. The focus of topological maps is how different nodes, representing points of interest in the environment, are connected to each other rather than the detailed geometric properties of the space. As such, these representations are usually in the form of graphs composed of nodes with edges representing the interconnectivity of nodes, an example of which is given in Figure 9. The edges in a topological representation can be assigned semantic properties, such as a cost of traversal or directionality [57]. Topological maps are often contrasted with geometric maps, although geometric maps can be and often are decomposed into topological representations. This was the case in [40,42], the authors of which considered an initial geometric map representation and used boustrophedon cell division, which is described in greater detail in Section 5.2.1. The result of boustrophedon cell divisions is a set of connected cells in the environment that takes the form of a topological representation. An occupancy grid (discussed in Section 4.4) was used by Kim et al. [44] to generate "waypoints" to ensure sensor coverage of the environment. These tasks can then be viewed as nodes in a graph connected by edges. Topological maps are rarely considered a priori knowledge; instead, another representation is decomposed into a topological map, as in [44]. Due to their simplified and abstract nature, they are better-suited to global path planning, with the specifics of path planning abstracted to an edge cost value.

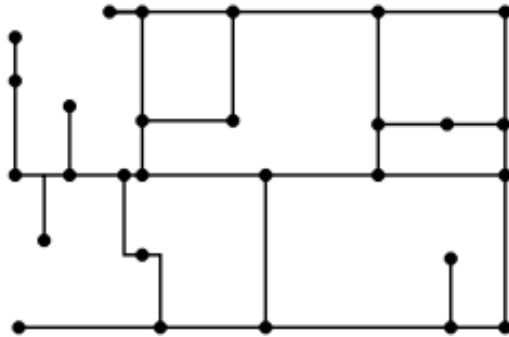

**Figure 9.** Topological representation [56].

### 4.3. Grid Map

A grid map is a grid of specified dimensions composed of squares of a certain size. Sometimes, the size of the grid cells represents the size of the robot's footprint or a sensor's footprint such that visiting each cell would provide full coverage of the environment [45,48,49]. Other times, the grid cell is used to discretise the possible positions [50] or to facilitate allocation of the environment to team members while still requiring a coverage path inside the cell [47]. In contrast to topological maps, with known dimensions and directional relations between cells, grid maps provide an abstracted yet accurate modelling of the environment's geometry. However, assuming that each cell represents a task, a grid map can be considered both a metric map and a topological graph. Kalde et al. [45] provided an example of encapsulating semantics in their grid map through the use of cell states. This can be seen in Figure 10a. In their work, the cells were in one of four states: unknown cells, denoted by question marks, represent those that have yet to be explored; occupied cells, denoted as black cells, represent static obstacles; animated cells represent robots, e.g., R1, and humans, e.g., H1; and free cells represent explored empty cells, denoted in white. Sharma et al. [48] used a similar representation. The model used in [49] can also be described as a grid map. Perez-imaz et al. [47] made use of a hexagonal grid map rather than a square one. Hexagonal grids facilitate diagonal movement with uniform distance between cells, as well as better approximation of a circular sensor radius than a

square, allowing an environment to be represented with fewer cells. A visualisation of the hexagonal grid map used by Perez-imaz et al. is shown in Figure 10b. Song et al. [46] made use of a multi-resolution grid-based environmental model. At the smallest resolution, the cells are the size of the sensor radius, and above that is a map of supercells composed of four cells. This multi-resolution grid is used to facilitate vehicles escaping local minima. While these simple multi-state grids are sufficient for the authors' uses, they do not take into account sensor uncertainty as per the aforementioned three main challenges [39] and are therefore unsuitable for real-world applications alone. However, such representations are compact and therefore particularly useful for high-level planning.

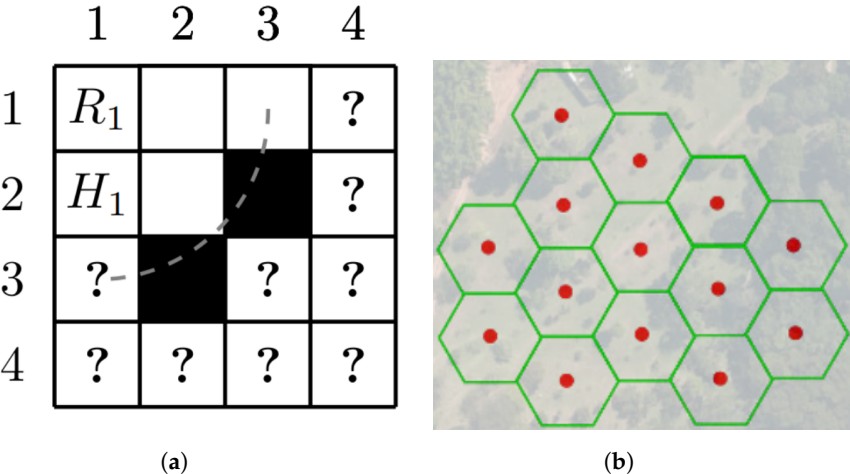

(**a**)  (**b**)

**Figure 10.** Comparison of grid maps. (**a**) Four-state grid map [45]; (**b**) Hexagonal grid map [47].

### 4.4. Occupancy Grid

Occupancy grids are common models used to tackle the uncertainty inherent in sensors. First proposed by Moravec and Elfes [58], the grid is composed of cells with values representing the probability of their occupancy by an obstacle. These cell occupancy probabilities were estimated as independent random variables which, while rarely—if ever—the case in the real world, simplify computation. Numerous approaches now exist to relax the assumption of cell independence [59]. Colares and Chaimowicz [51] mad use of an occupancy map in their approach to exploration. The occupancy map was generated based on their SLAM approach. In their approach, the occupancy grid is used to compute the costs of frontiers for the team members. Bramblett et al. [52] also mad use of an occupancy grid representation, using recursive Bayesian estimation to update the cells given sensor measurements. In this case, the occupancy is once again used to identify frontier cells, and exploration tasks are generated in areas of high uncertainty, facilitating complete sensor coverage of the environment. Occupancy grid representations are particularly useful in unknown environments, as they require no prior knowledge for their formation. In regard to OWT inspection, occupancy grids have three main areas of use. By forgoing the assumption of prior knowledge, occupancy grids can account for sensor uncertainty while facilitating the mapping of an unknown environment. Occupancy values can act as a component in an objective function to drive the team to explore uncertain areas. Occupancy grids can also be used to construct a cost map (see Section 4.6) for motion planning, providing a tradeoff between traversing unknown areas and distance. The effect of the aforementioned cost map is that a robot would have a degree of reluctance to traverse unknown areas due to potential of encountering obstacles or dead ends.

### 4.5. Voxel-Based Mapping

A voxel is a cell in a 3D grid, the term voxel being "an analogy to pixel" [60]. Voxel-based mapping represents the environment as a 3D grid composed of voxels. The simplest voxel representation is a 3D binary array, with 1 representing occupancy and 0 repre-

senting free space [61]. Two implementations of voxel-based mapping were reviewed: Octomap [53,54] and a Euclidean signed distance field (ESDF) map [55]. Octomap is a probabilistic framework for environmental modelling of 3D cell occupancy based on hierarchical octrees [62]. The hierarchical nature of the approach can reduce memory usage and facilitate varying levels of environmental detail to be captured. Areas with fewer features or in which mapping is less critical can be mapped at a lower resolution, conserving memory and reducing the computational cost of future environmental decomposition. Conversely, those areas of particular interest can benefit from a higher resolution, allowing for more detailed and accurate mapping and better-informed environmental decomposition. Dong et al. [54] made use of an Octomap for their exploratory scanning before decomposing it into a 2D occupancy grid to plan tasks. Dornhege et al. [53] provided a task-planning algorithm working directly with an Octomap. The use of Octomaps is a very powerful approach to modelling 3D environments, and its availability as an ROS library has added to its popularity. An ESDF was used by Bartolomei et al. [55] and described initially in [63]. This is a highly semantic voxel model in which each voxel is linked to a data structure composed of the voxel's coordinates, the occupancy probability, the Euclidean distance to the nearest obstacle, whether the voxel has been observed, the voxels closest to itself, and information on the area surrounding its closest obstacle. The authors described the approach's usefulness in UAV navigation, as "what is truly useful is the information of free space, instead of obstacles". The difference between the occupancy grid and ESDF model is shown visually in Figure 11. Voxel-based mapping approaches are, in their regular dimensions and regular directional relations (each voxel shares the same spatial relationship with their six neighbouring voxels), similar to the 2D grid-based representations discussed previously. However, unlike a 2D grid map, they have little use outside of 3D coverage. While 2D coverage often involves visiting each cell in the environment once, as in [49], this is rarely the case in 3D coverage. Rather, 3D coverage tasks tend to involve sensor coverage of either the whole environment [55] or subsections of the environment of particular interest [53]. Arguably, these tasks are particularly representative of offshore wind inspection, especially in the case of covering specific areas of interest within a 3D environment, which can represent turbines themselves or areas of the turbine of special interest, such as the blades.

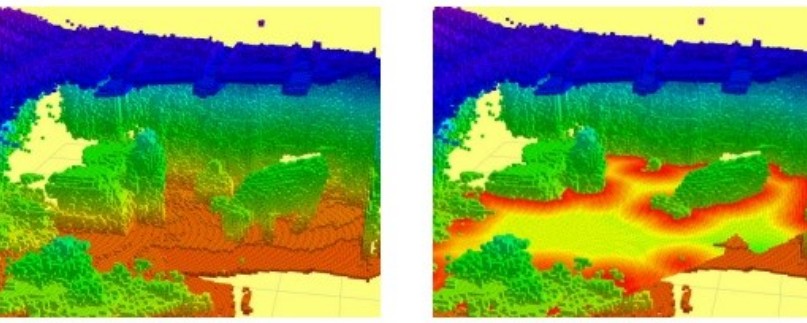

**Figure 11.** An occupancy grid model (**left**) and an ESDF model (**right**) [63].

*4.6. Cost Maps*

Cost maps are grid-based representations in which the value of each cell expresses a cost of traversal. The cost of a cell, specified as a numerical value, can represent a number of different attributes of traversing a given cell. In the work of Ball et al. [40], the attribute in question was a deviation from a high-level planned path to avoid obstacles. They then made use of a search-based lattice planner (SBLP) to generate paths that minimise the cost of traversal with respect to the high-level path and detected obstacles. Although not expressed explicitly, the ROS navigation stack uses a 2D cost map, so other works making use of ROS are very likely to make use of them also [64]. The ROS cost map builds an occupancy grid and, based on the occupancy value of a cell, increases the cost of traversal within a user-specified radius around the suspected obstacles. The effect of this is that the

path-planning algorithms account for the cost map and select paths based on a tradeoff between distance and proximity to suspected obstacles. Regarding the coverage path-planning problem, cost maps find their greatest utility in motion planning, facilitating the generation of paths between tasks while trading off between the potential of encountering obstacles and the distance to travel.

*4.7. Discussion*

A wind farm can be considered an extreme, sparse, and unstructured environment. The environmental model relates to the task being undertaken, i.e., OWT inspection, but there are variations in this task. The environmental modelling approaches one should select depend on a variety of aspects. These aspects can include whether a model is known a priori, if the team is homogeneous, and whether the blades are moving. This discussion attempts to map the suitability of the environmental model taxonomic classes to the inspection problem, keeping in mind the aforementioned characteristics of the environment (extreme, sparse, and unstructured). The inspection of a turbine involves acquiring sensor data across the entire turbine or in specific areas of interest such as the blades. Figure 2 provides a basic description of the components of an OWT. This variant of an OWT inspection is a 3D structural inspection, a variation of coverage in which the aim is to ensure sensor coverage of either the entirety of a structure of interest or specific areas of said structure. The 3D nature of this task excludes the use of 2D environmental models, lending itself to a voxel-based model.

Considering inspection as the coverage of the structure, in a standard, single-resolution voxel-based model, the rest of the environment is modelled in the same detail as the turbine. The result of this single-resolution voxel-based model is inefficient memory use and slower computation of task decomposition and motion planning (Task decomposition and motion planning are discussed in Section 5.2.1 and Section 5.2.2, respectively). Therefore, there is an incentive to make use of an adaptive resolution like that provided by an Octomap [62]. In doing so, the turbine can be captured in a detailed and accurate voxel-based representation without also requiring a detailed model of the empty space around it. An additional benefit of an adaptive resolution for OWT inspection is that it allows for varying levels of coverage detail based on the specific turbine component being inspected.

One aspect of OWT inspection that may require a novel solution not seen in the reviewed works is the inspection of moving blades. Blade inspection generally requires the turbine to stop, but there is a financial incentive to keep the turbine running during inspection. While an Octomap is updatable and can represent dynamic environments, there are not any semantic labels attached to voxels to represent which blade is which—just a value to specify the probability that the voxel is occupied. To ensure coverage of all the blades, the environmental model needs to keep track of which blade is which. This could be achieved by applying a semantic label to the moving cluster of voxels that represents an individual blade; however, this presents challenges such as keeping track of which blade is which when not in view. In the reviewed literature, no modelling approach accounts for these moving tasks, representing an avenue for future research. An unaddressed area is representing heterogeneous tasks, as inspection tasks may require more than one class of robot. Different types of tasks or motion capabilities in a heterogeneous team need to be represented in the environmental model. These requirements of certain capabilities could be represented semantically in topological models by labelling edges based on traversal requirements or task nodes with information on the necessary capabilities. In order to semantically label the edges with these traversability requirements, a novel form of heterogeneous traversability analysis needs to be implemented, which is an open area of research.

## 5. Decision Making

This section aims to provide an answer to subquestion 2 from Table 2: *What is the most suitable multi-robot coverage decision making approach for domain applications resembling*

*offshore wind inspection?* In this work, as mentioned previously, decision making, as defined by the authors of this paper, is the mechanism by which collective choices are defined by centralised or decentralised objectives. In order to systematically analyse the approaches' suitability for the domain, two taxonomies were applied. The first taxonomy relates to the amount of knowledge available to a system a priori. Almadhoun et al. [33], in their survey on coverage path planning, classified approaches assuming prior knowledge as being "model-based" and those without prior knowledge as being "non-model-based". Model-based approaches assume a prior environmental model, i.e., a "known environment", whereas non-model-based approaches lack this initial knowledge. Knowledge of one's environment is a significant advantage, and as one would expect, model-based approaches usually achieve better performance. Prior knowledge of one's environment is a strong assumption, and this prior knowledge is not always available or accurate. The second taxonomy uses the planning definition proposed by Yan et al. to analyse the works included in this review [36]. Yan et al. consider planning to be composed of two aspects: task planning and motion planning. Task planning can be further divided into two subaspects: task decomposition and task allocation, which are concerned with turning a multi-robot task into a set of single-robot tasks and then allocating these tasks to the team. Motion planning involves the generation of paths and trajectories for the team members to travel to and complete all the tasks.

### 5.1. A Priori Knowledge

Almadhoun et al. [33] identified a dichotomy in approaches to coverage. Approaches can either have or not have an a priori environmental model. The authors defined these groups of approaches as either non-model-based or model-based.

### 5.1.1. Non-Model-Based Approaches

In the simplest sense, a non-model approach to coverage assumes nothing about the structural environment, facilitating coverage without requiring a prior environmental representation. Therefore, these approaches are often used when the environment is unknown or uncertain. Non-model-based approaches can be described using the terms "exploration" [50] and "coverage of an unknown environment" [46]. There is a degree of ambiguity in the terms "coverage" and "exploration". Yamauchi [65] defined exploration as a problem of, "Given what you know about the world, where should you move to gain as much new information as possible?". A commonality among papers concentrating on the exploration problem is that the approaches attempt to maximise knowledge of an a priori unknown environment, that is to say, exploration aims to model the environment and work to maximise the completeness of the model. On the other hand, coverage can be roughly split into two distinct problems: (1) covering an environment with the footprint of a team of sensors in an optimal manner and (2) assigning spatially distributed tasks to a team of robots in an optimal manner. The former is often decomposed into the latter, and the latter is an instance of the multi-robot task allocation problem [66]. Coverage can be considered in an unknown environment without exploration. In [52], a team of robots with a limited communication range were tasked with exploring an unknown environment. The authors considered an unknown environment with tasks; hence, the problem required both optimal full exploration and task allocation coverage. "Exploring an environment by repeatedly applying path planning algorithm at each instance of time" is a highly specific definition of exploration proposed by Sharma et al. [48], characterising the online (Online in the sense of planning for a robot indicates that the plan is generated during run time, whereas offline plans are generated before the execution) nature of the exploration problem. The quality of sensor coverage was taken into account in the work of Dong et al. [54], who stated their problem was collaboratively exploring and mapping a scene such that scanning coverage and reconstruction quality were maximised, while the scanning effort was minimised. Among the reviewed works, eight involved non-model-based approaches .

5.1.2. Model-Based Approaches

An approach can be said to be model-based if it assumes a prior environmental model, an assumption that simplifies task decomposition [33]. Ball et al. assumed a known geometric map using multiple modified John Deere TE Gators for crop spraying [40]. This geometric map was then decomposed into multiple subregions through boustrophedon cellular decomposition. An approach that initially assumes environmental bounds provided by a set of vertices representing the bounds of an area of interest, which is then decomposed into a hexagonal grid was proposed by Perez-imaz et al. [47]. Karapetyan et al. considered a known geometric model that is then decomposed into task areas via boustrophedon cellular decomposition [42]. Kim et al. [44] also assumed a search region of an arbitrary shape. Zhang et al. assumed a prior model of the environment in the form of a simple binary grid map of free cells and obstacles, which is a very common representation in offline coverage problems [49]. Finally, Tang et al. considered a known geometric model of the environment. However, due to random dynamic interference, their approach cannot be computed offline [43]. A prior model of the environment can facilitate prior planning and optimal solutions to task decomposition and path-planning problems. However, solutions that rely too heavily on prior knowledge of the environment may struggle with the uncertainty of a real-world implementation, especially in areas with high uncertainty, like offshore wind farms. Eight of the reviewed works involved model-based approaches.

*5.2. Planning*

Planning is defined as "the task of coming up with a sequence of actions that will achieve a goal" by Yan et al. [36]. Planning for a mobile multi-robot system can be divided into task planning and motion planning. Task planning is a problem of how tasks should be divided among the team, while motion planning is concerned with devising paths in order to facilitate locomotion to and completion of said tasks.

5.2.1. Task Planning

Yan et al. [36] defined Task planning as the problem of "which robot should execute which task". They then proposed splitting task planning into two further categories: task decomposition and task allocation. Task decomposition concerns how a multi-robot problem can be split into single-robot tasks, and task allocation involves how best to assign these single-robot tasks to the team of robots. The works reviewed in this study are grouped according to the task decomposition method as detailed in Table 7.

**Table 7.** Task-planning approaches.

| Paper | Task Decomposition | Task Allocation |
|---|---|---|
| Ball et al. (2015) [40] | Boustrophedon cell division | Not described |
| Kalde et al. (2015) [45] | Frontiers and humans identified as potential tasks | Greedy allocation on a cost matrix |
| Song et al. (2015) [46] | Pre-decomposed | Pre-allocated |
| Colares and Chaimowicz (2016) [51] | All frontier cells as tasks | Optimal frontier based on a cost function |
| Dornhege et al. (2016) [53] | Set of optimal views | Greedy allocation or set the coverage solution with TFD solver |
| Perez-Imaz et al. (2016) [47] | Hexagonal grid | K-means clustering |
| Sharma et al. (2016) [48] | Pre-decomposed | Pre-allocated |
| Masehian et al. (2017) [41] | Hierarchy of decompositions | Allocated based on other classes of robots identifying tasks |
| Karapetyan et al. (2018) [42] | Boustrophedon cell division or DCS path splitting | Not described |
| Dong et al. (2019) [54] | Set of optimal frontier views | K-means clustering |

**Table 7.** *Cont.*

| Paper | Task Decomposition | Task Allocation |
|---|---|---|
| Zhang et al. (2019) [49] | DARP | DARP |
| Bramblett et al. (2022) [52] | All frontier cells as tasks for exploration; tasks discovered in exploration | K-means clustering, auctioning, and optimal frontier based on a cost function |
| Kim et al. (2022) [44] | Frontier cells based on the uncertainty of neighbours | Heterogeneous k-means clustering |
| Tang et al. (2022) [43] | N/A | N/A |
| Bartolomei et al. (2023) [55] | Exploration: clustered frontiers [67]; Collection: uncovered trails | Exploration: optimal frontier-based with minimal cost; Optimal trail-based with minimal cost |
| Yu et al. (2023) [50] | N/A | N/A |

5.2.1.1. Area Decomposition

The works discussed in this section share in common the decomposition of a 2D plain into a set of geometric shapes representing coverage areas to be assigned.

In [40], the initial representation was in the form of a geometric map. The task was decomposed using boustrophedon cell decomposition, as first described in Choset's work [68]. The boustrophedon cell decomposition algorithm takes a known geometric model and decomposes it into a topological representation composed of uneven cells based on the model's geometry. This approach works by running a vertical line along the geometric model, and when an obstacle bisects line two, the current cell is closed, and two new cells are created as shown in Figure 12. The result of this is several cells that can be covered in a boustrophedon motion (back and forth). The resulting cells are allocated to the team of robots, but the details of this were not provided.

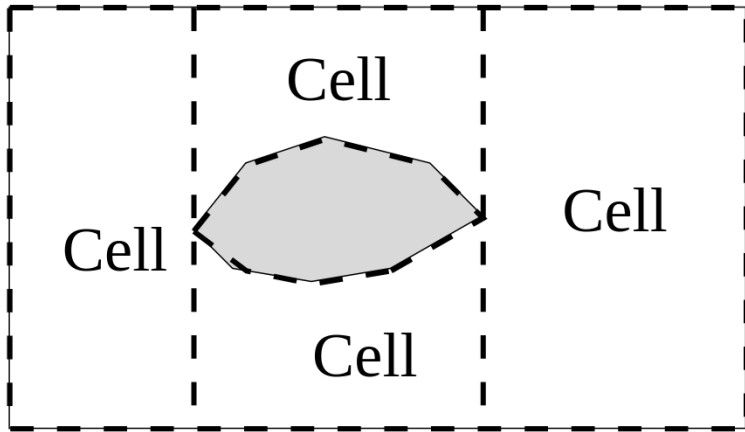

**Figure 12.** Boustrophedon cell decomposition [68].

UAV coverage for first-response rescue and recovery with UAVs was implemented by Perez-imaz et al. [47]. Hexagonal decomposition was used to decompose the task, which worked by overlaying the hexagon over the known geometric environment, with the hexagon's size representing the sensor range. The tasks are allocated using K-means clustering, and each hexagon within a Graph is formed. While the approach considers a multi-robot team, the conducted real-world experiments carried only used a single UAV.

A purely offline approach was considered by Karapetyan et al. [42] in their approach to autonomous surface vehicle coverage. Their method adopts two approaches to environmental decomposition: boustrophedon cell division, as used by Ball et al. [40], and a Dubins coverage solver (DCS). The DCS splits the environment into several passes to form a graph, outputting a Hamiltonian path. In Dubins coverage with route clustering, this Hamiltonian

path is then split among the team. Another approach known as Dubins coverage with area clustering segments the environment with Boustrophedon cell division, clusters cells together, then uses the DSC to create the tasks. Task allocation was not discussed.

All the works reviewed in this section consider the coverage footprint and the sensing platform to be inseparable, i.e., an individual robot has a sensing footprint of a specified size centred on itself. In a 3D structural inspection, this is not the case; rather, the sensor footprint is always separate from the sensing platform. While the relevance of decomposing a 2D space into several regions to be covered does not have an obvious application in OWT inspection, it is worth considering the extendability of the reviewed approaches to the 3D structural inspection problem. One possible avenue for this is the decomposition of 3D space into a set of assignable regions. Both DARP and boustrophedon cell divisions rely on 2D geometry to decompose the environment, so extending them may not be simple. As the problem we are considering involves sensor coverage of a structure, segmenting the environment without consideration of the structure to be sensed would likely result in suboptimal solutions. An alternative approach is to use these task decompositions as a component of a larger task decomposition approach. In the case of OWTs this could involve the use of an area segmentation method to decompose the structure's surface into continuous sections that can be assigned to the robots within the team. Following this, the coverage problem can be seen as a set of single-robot coverage problems in the assigned regions.

5.2.1.2. Frontier-Based Decomposition

The concept of frontier-based exploration was first introduced by Yamauchi [65] in 1997. These approaches harness environmental uncertainty to generate tasks or viewpoints iteratively, allowing for exploration or coverage with limited knowledge of the environment. Viewpoints are usually selected based on a cost function aiming to maximise the reduction in uncertainty upon moving to them.

Kalde et al. [45] considered the problem of exploring an unknown environment with wheeled robots. The authors' approach to this problem is frontier-based iterative planning with human guidance. At each planning interval, the environment is first decomposed by identifying tasks as either frontier tasks or human tasks. In this work, humans can assist robots in navigation by leading them. The approach makes use of a parametric heuristic to equilibrate the frontier tasks and the human tasks. This parametric heuristic takes the form of a "mixed cost model", where a cost value is computed for each agent–task assignment in a cost matrix. The cost function is formed from two components. The distance component is simply the distance for the robot to traverse to a task. The penalty component is composed of a time penalty and an orientation penalty, the time penalty being the time elapsed since the frontier's discovery and the orientation being the smallest angle between the robot's orientation and the direction of the frontier or orientation of the human. Given the cost matrix, two greedy approaches were used—one fully decentralised and one locally coordinated.

Colares and Chaimowicz [51] considered an instance of the exploration problem using a frontier-based approach. The task was decomposed by considering all known frontier cells' potential targets. For task allocation, a three-component cost factor was used in a distributed fashion. The first component is an "information factor", which quantifies the potential information gained by visiting a cell based on its neighbours. A distance component was used, with two variables to change the behaviour by favouring close or distant frontiers. Finally, a coordination factor penalises selecting a frontier close to a known neighbouring robot. A visualisation of this frontier method is given in Figure 13. Therefore, the optimal frontier is selected for each robot. The approach was successfully implemented with two Pioneer 3AT wheeled robots to explore an indoor environment.

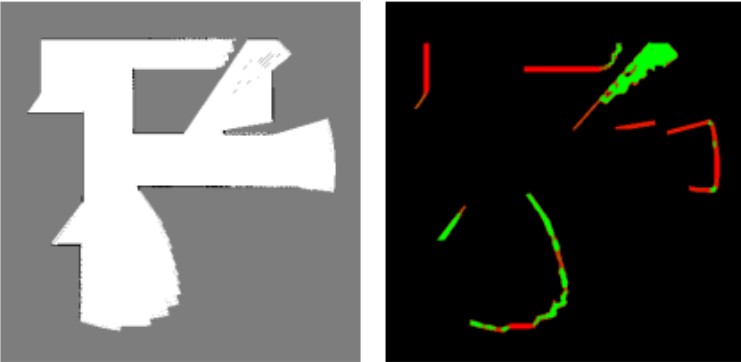

**Figure 13.** Colares & Chaimowicz's frontier detection [51].

Another exploration approach was considered by Dong et al. [54]. This was implemented in an indoor environment with a team of up to six Turtlebots. The authors considered an Octomap representation projected on the floor plane and used the uncertainty to decompose the task. The approach also uses a validity map, which provides the possible poses of a sensor. The voxels positioned on frontiers are sorted according to uncertainty in a priority queue; then, the validity maps are considered to find poses that rays pass through the voxel. The pose with optimal validity is selected, where validity is composed of the deviation from 0 degrees and a function of the optimal distance. For the view being selected, all voxels within its view are removed from the queue. This process repeats until a specified number of views is generated. As for task assignment, the problem is viewed as an optimal mass transport problem. This problem is formulated and then discretised to an objective function with three components to be minimised. A compactness component penalises spatial scattering of assigned tasks, a distance component minimises travelling cost, and a capacity component ensures robots can complete only some tasks within their capacity for a given interval. This was then optimised using a modified k-means clustering algorithm. Zhang et al. [49] considered area coverage using UAVs with mobile charging stations. The continuous area was initially split into tasks through gridmap decomposition. They then made use of a modified version of the DARP (Divide Areas based on Robot's Initial Positions) algorithm to avoid discontinuities via edge detection. This effectively allocates areas of the grid for coverage. The authors made use of Crazyflie UAVs adapted for mobile charging and wheeled mobile charging stations.

As discussed previously, Bramblett et al. [52] were concerned with the problem of coverage of tasks in an unknown environment; as such, planning has to occur for both tasks. For exploration, a Sobel operator is used on the occupancy grid to identify frontiers based on the gradient between known and unknown space. Those frontiers representing obstacles are discarded. Naive to and in tandem with edge detection, the unknown environment is clustered using K-means clustering for each robot. Tasks are then auctioned to the robots in a centralised manner. The robots then act in a greedy manner using a cost function that favours closer tasks, but tasks outside of the robot's assigned task area are penalised according to their distance from the task area. Regarding the coverage aspect, the tasks are "decomposed" from the environment in the sense that they are discovered during exploration. The decision logic for coverage is shown in Figure 14. This search involves seeking a robot that does not have a rendezvous, which indicates that it is likely to have found a task. Exploitation is the act of working on a task. The authors implemented the approaches using three Husarion ROSbot 2.0 UGVs.

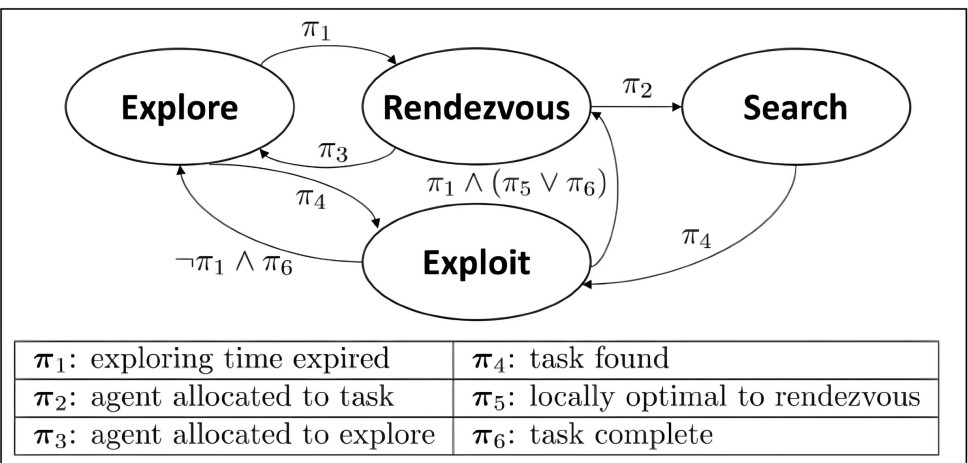

**Figure 14.** Bramblett decision logic [52].

Kim et al. started with a geometric representation of the environment [44]. Their approach considers the degree of heterogeneity in the sense that team members have different sensor ranges. The tasks are generated based on the smallest sensor range while grouping unknown frontier cells. Task assignment is treated as a clustering problem. The authors extended K-means clustering to their Heterogeneous clustering algorithm. This clustering algorithm considers both the spatial proximity of two agents and the weighted distance based on the sensing ability of the specific robot.

Finally, Bartolomei et al. discussed the exploration of forests with a team of UAVs [55]. This method involves two modes for the robots in the team: exploration and collection. Exploration is, as expected, focused on obtaining new knowledge of the environment, while collection is concerned with cleaning up the "trails" of unexplored areas left by exploration. Exploration tasks are decomposed from clustered frontiers. The authors did not specify the clustering algorithm they used. Given the cluster centroid, candidate views in a cylinder focused on the centroid are considered. The view with the highest coverage of the cluster is then selected as the optimal view for that cluster. A set of optimal views among the clusters from the decomposed tasks is used for task allocation. These clusters also undergo classification, with those representing trails being semantically classified as such based on their isolation. A mechanism for declaring areas of interest for a team member takes place between two robots if they are inside the communication range. This area of interest is used by the robot to select tasks from the previously discussed set of tasks. The exploration assignment is based on a cost function with four components: distance and change-of-direction components, a label component to penalise trails, and a component to valorise views near the area of interest. As for those team members assigned as collectors, they cover trails through a cost function only considering distance and proximity to the area of interest.

The value of frontier-based approaches is their coverage application when knowledge of the environment is limited and there are uncertainties in regions of the map. When considering the applicability of these methods to the OWT inspection problem, it is important to take into account prior assumptions. If the entire environment is considered known a priori, frontier-based approaches have few clear benefits. Frontier-based approaches can work in a decentralised manner, potentially performing better where communication may be limited. In situations where the turbine's location may be uncertain, such as in cases involving floating turbines in areas with large currents, there could be value in using a frontier-based approach, ideally while still accounting for the known general geometry of the OWTs.

5.2.1.3. View Decomposition

While there is only one example of this decomposition approach, it proves to be one of the most applicable. View decomposition attempts to, for a known structure in a known location, find a set of views that optimally cover the surface of the said structure with the sensor footprint

Dornhege et al. [53] tackled a coverage search problem with a team of wheeled robots. The authors considered an Octomap environmental representation with a known search set of voxels. For each voxel in the search set, several random vectors are generated; then, ray tracing is used to find a set of grid cells that represent possible sensor states along the vector. The corresponding grid cells from the ray tracing are used to increment a utility function for the grid cells, which is done for all cells in the search set to create a utility map across all accessible sensor states. States over a given utility threshold are added to a set of useful sensor states. For task allocation, the problem can then be considered a set cover problem, given a search set and a set of sets representing those observers' cells for a sensor position, finding the minimal set of sensor positions that cover the environment. Dornhege et al. used a variant of the temporal fast downward planner to solve this set cover problem as a planning problem. Alternatively, the authors used a greedy approach in which the views were selected for each robot iteratively based on the cost. The cost in the greedy approach can either balance the view utility and the travel time or be the travel time. This task allocation method also ensures a high-level path plan.

While this approach may be the most readily applicable to the OWT inspection problem, it would need modification for this use case. The authors considered only wheeled robots and chose the possible sensor states based on this assumption; for OWT inspection, USVs and UAVs would be necessary. USV sensor positions have specifications similar to those of wheeled robots, as they are limited to the surface level, whereas UAVs are not bound to the surface. The capability of UAVs to reach almost any position in space would possibly make the approach proposed by Dornhege et al. [53] infeasible owing to computational complexity. The authors also did not account for camera orientation and distance concerning the surface of the structure to be inspected. The application of a method similar to theirs to OWT inspection would need to account for sensing quality by ensuring that the sensor is positioned to capture useful information, which can be achieved by requiring a certain proximity and orientation relative to the surface being captured. Applying these requirements would benefit the computational complexity of the solution by reducing the number of possible sensor states to those that fulfil the requirements. This approach also does not account for uncertainty in the environment, requiring a full a priori model with no dynamics.

5.2.1.4. Hierarchical Decomposition

Masehian et al. [41] proposed an interesting hierarchical, heterogeneous approach to coverage of an environment with limited sensing capabilities. In their approach, there are three classes of robots, each with a different sensing capability and differing behaviours. A quadridirectional robot with four quadridirectional sensors is not assigned tasks as such; rather, it initially starts a boustrophedon motion across the environment. The quadridirectional robot identifies obstacle and wall boundaries, which represents task decomposition for the second robot class, i.e., a boundary-following robot equipped with a radial sensor. The assignment for a boundary-following robot class is the robot with minimal distance. This robot follows the boundary, and if a sensed point does not align with the last two points, a task is created for the last robot in the hierarchy. The gap robot can identify gaps between obstacles within a radius and is therefore assigned the potential gaps identified by the boundary follower.

While this work addresses a very specific case, it touches on an interesting aspect of the OWT inspection problem. When addressing OWT inspection, there is potential to utilise the heterogeneity of capabilities to increase the quality of inspection. While this is not necessary for the problem as defined by us, it could be of practical use in industry,

acting as a variant of the inspection problem. In other words, identifying areas of interest would involve identifying a damaged area from a distance with a suitable sensor, and then using a team member with a different sensor to elaborate on the identified damage by moving closer. This behaviour, while not identical, resembles the approach used by Masehian et al. [41] in the generation of a hierachy of tasks based on the sensing capabilities of other team members.

5.2.1.5. Pre-Decomposed and Non-Decomposing Approaches

Some works reviewed is this study did not consider the decomposition of the environment, and others, such as those involving reinforcement learning approaches, did not view task decomposition as a problem separate from motion planning; as such, the specifics of these works are discussed in greater detail in the motion planning section.

Song et al. [46] focused on the use of AUVs for full sensor coverage. In their approach, the environment is assumed to be pre-decomposed into subregions, and initial task allocation is not considered. Instead, their approach focuses on motion planning and fault tolerance, the former of which is discussed in Section 5.2.2 and the latter of which is discussed in Section 6.

Sharma et al. proposed an approach in which both the environment is pre-decomposed and task areas are pre-assigned [48].

Tang et al. [43] proposed a worker station approach to coverage. In this approach, the environment is decomposed into a grid, but the resulting cells cannot be viewed as tasks. The authors used a reinforcement learning approach, so tasks were not allocated as such. The reinforcement learning approach's action space is concerned with the linear and angular velocity of a single robot, as discussed in Section 5.2.2. The approach was implemented using a skid-steer wheeled robot as a station and two differential-driven wheeled robots as workers.

5.2.2. Motion Planning

As previously described, motion planning is concerned with devising paths to facilitate locomotion to and completion of the previously planned tasks. In some cases, motion planning alone is used without task allocation; for example, greedily covering a geometric map may not involve discrete tasks but only motion planning. Path planning is defined by Kavraki and LeValle [69] as finding a collision-free path from an initial pose to a goal pose. The problem being considered here more closely resembles the multi-goal path-planning problem proposed by Wurll et al. [70], i.e., finding a collision-free path connecting a set of goal poses while minimising a cost function. Solutions often consist of two tiers: a global tier and a local planning tier. Global planning approaches solve the multi-goal path-planning problem at a higher level, sometimes forgoing consideration of collision altogether, while local planning more closely resembles the traditional path-planning problem, concerned with a path from an origin to a goal while avoiding collision and minimising cost. The approaches in the reviewed works are summarised in Table 8.

**Table 8.** Summary of motion planning approaches.

| Paper | Motion Planning Approach |
| --- | --- |
| Ball et al. (2015) [40] | Search-based lattice planner with a local pure pursuit controller |
| Kalde et al. (2015) [45] | Potential field on a grid map |
| Song et al. (2015) [46] | Generalised Ising model with local and global navigation mechanisms |
| Colares and Chaimowicz (2016) [51] | Not specified beyond iterative task selection |

**Table 8.** *Cont.*

| Paper | Motion Planning Approach |
| --- | --- |
| Dornhege et al. (2016) [53] | Single TSP problem solved with Temporal Fast Downward planner or Lin–Kernighan heuristic or a single greedy approach split according to the number of robots |
| Perez-Imaz et al. (2016) [47] | Dijkstra's algorithm on a hexagonal graph with a lawnmower pattern |
| Sharma et al. (2016) [48] | Directional motion and nature-inspired algorithms |
| Masehian et al. (2017) [41] | Different policies for different classes of robot: boustrophedon motion, boundary following, and guide-path following with obstacle avoidance |
| Karapetyan et al. (2018) [42] | Dubins coverage solver with TSP problem solving |
| Dong et al. (2019) [54] | Christofides' algorithm for TSP approximation with path smoothing |
| Zhang et al. (2019) [49] | Spanning tree coverage algorithm |
| Bramblett et al. (2022) [52] | A* path-planning algorithm with iterative frontier-based tasks |
| Kim et al. (2022) [44] | Genetic algorithm for TSP problem with A* algorithm and B-spline for path computation |
| Tang et al. (2022) [43] | Reinforcement learning with a multi-layer perception for the policy network, with the action space comprising angular and linear velocity |
| Bartolomei et al. (2023) [55] | Trajectory generation integrated with task allocation |
| Yu et al. (2023) [50] | Reinforcement learning with a decentralised multi-tower CNN-based policy. The action space represents a global goal, and local navigation is achieved with the A* algorithm. |

An example of this distinction between global and local path planning was provided by Ball et al. [40]. In this approach, the global planner makes use of a search-based lattice planner to find the best path, both considering the cost of motion primitives and minimising the cost of traversing a cost map while avoiding obstacles. A local pure pursuit controller is used for the global planner path if followed optimally, using two PI controllers to minimise the error in the robot position and the global planner path. If a collision is detected in the global path, the local pure pursuit controller can reject it and ensure that the global planner recomputes a new path. Kalde et al. [45] described motion planning using a potential field propagated on the grid map. Another two-level approach to motion planning was considered by Song et al. [46]. As discussed earlier, the authors considered a multi-layer grid representation, and their local navigation works on the lowest level of this grid. They described their navigation as being based on a generalised Ising model. The cells within the Ising model can be categorised as one of three states: obstacle, explored, or unknown. Local potential energy is formed based on the state of the cell and its neighbours. A component of the local potential energy is a constant potential energy field that encourages back-and-forth motion for coverage. For each robot, the target is therefore the cell with the highest energy potential. But it may be the case that a robot could get caught in a local minimum. The authors accounted for this eventuality with a global navigation mechanism. Global navigation works on a coarser grid than local navigation, using a low-dimensional probability vector to restore environmental information for the coarser grid. Then, much as with local navigation, a target is selected and navigated towards until local navigation is possible. Colares and Chaimowicz's work does not discuss the specific motion planning approach implemented beyond the iterative task selection previously described [51]. Dornhege et al. approach path planning by treating it as a set of single

travelling salesman problems (TSPs) (The travelling salesman problem is a well-known mathematical and computer science problem that can be summarised as, "Given a list of cities and the distances between each pair of cities, what is the shortest possible route that visits each city exactly once and returns to the origin city?"), given the result of their set cover problem. The authors solved the TSP problem for each subset using either a Temporal Fast Downward planner [71] or a Lin–Kernighan heuristic [72]. An alternative method the authors used involves extending the single-robot greedy allocation by taking a single greedy plan for the environment and splitting this path according to the number of robots. These approaches provide a global path plan for traversing a topological graph, but details on path planning to account for the structure of the environment itself are sparse. Given the topological graphs decomposed and clustered from the hexagonal graph presented by Perez Imaz et al. [47], the authors ensured a lawnmower pattern within the hexagons by using parallel lines intersecting with the hexagon to create nodes, then using Dijkstra's algorithm to generate the path. An optimal angle of the path for each hexagon is found to minimise the complete coverage distance. Sharma et al. [48] split the environment into several task areas to be covered. Until the entire areas for a robot are covered, an iterative path-planning approach is adopted. At any given iteration, the robot randomly chooses one of two motion policies: directional motion or a nature-inspired algorithm. Directional motion has two variations: a directional scattering effect moves in the direction of a "cluster head" selected randomly to encourage exploration or, under a zigzag search effect, the cluster head is chosen dynamically, ensuring more random and less directional motion. The authors modified particle swarm optimisation, bacteria foraging, and bat algorithms for multi-robot exploration. The particle swarm optimisation algorithms were found to perform optimally for exploration. The motion planning described by Masehian et al. [41] takes into account the heterogeneous nature of the team involved. Each of the three classes of robot has a different motion planning policy. The quadridirectional robots use boustrophedon motion to cover the task area. During boustrophedon motion, the robot may become trapped in a corner, which is resolved by referring to its observation history and finding any gaps it may have passed since its last row, backtracking to this point and continuing in the direction of that gap. The boundary-following robot follows the boundary of an obstacle or wall. This is done by considering a band representing an optimal distance from the obstacle around the edge of said obstacle that the robot should stay within. The gap robot's motion planning can be considered a classical path-planning problem; given a task, the aim is to find the optimal path from the current location to the path. The robot considers a guide path, which is the direct line from the robot's position to the task. Upon the gap robot sensing an obstacle along the guide path, it randomly decides whether to go left or right around the obstacle. Karapetyan et al. [42] used the Dubins coverage solver proposed by Lewis et al. [73], an approach that involves solving the TSP problem for generated rows while accounting for Dubins constraints. A TSP problem was also considered by Dong et al. [54] after selecting some tasks for a given robot. For this purpose, the authors used the Christofides algorithm to calculate a TSP approximation, the path of which was then smoothed. Examples of paths over an iteration are shown in Figure 15.

Zhang et al. [49], previously having decomposed the environment into a set of task areas, produced a coverage path common to offline coverage of gridmap environments a spanning tree coverage algorithm. Spanning tree coverage capitalises on the grid structure by grouping sets of four cells together and considering these supercells to find a spanning tree. This spanning tree can then be traversed by the robots in the team, forming a cycle across the entire environment. Bramblett et al. [52] considered the iterative frontier-based task as discussed earlier. To navigate to these tasks, they used the A* path-planning algorithm. Kim et al. [44] used a topological graph, as discussed earlier, and considered the TSP problem. Their approach to the TSP problem is a genetic algorithm [74]. The path between the points in the TSP solution is then computed using an A* algorithm, and a spline function of that using a B spline is implemented. Tang et al. [43] were concerned with the use of reinforcement learning coverage; as stated earlier, this doesn't involve task

planning as such. The authors described their multi-agent reinforcement learning problem as a "Decentralized Partially Observable Markov Decision Process". They followed a centralised training and decentralised execution paradigm. The observation space consists of the robot's information, information from its sensors, and information from robots within communication range. The action space consists of linear and angular velocities. Their reward function is composed of four components. A completion reward is awarded for finishing coverage, and the second component approximates worker capacity, providing a negative reward if the energy capacity of a robot is beneath a threshold value. During the training phase, robots can continue coverage with depleted capacity. The third reward is a negative reward for collisions, and the final component is a constant negative reward to encourage an optimal time. The authors used a multi-layer perception for their policy network. Path planning was computed by Bartolomei et al. [55] through task allocation, and the trajectories were then generated using the approach proposed by Zhou et al. [75]. Another reinforcement learning approach was considered by Yu et al. [50]. For this purpose, the authors made use of an asynchronous variation of the multi-agent proximal policy optimisation algorithm [76]. The task is modelled as a decentralised, partially observable semi-Markov decision process. A multi-tower CNN-based policy is used for each agent. The action space is a global goal, but atomic actions enact this goal using the A* algorithm. A three-component reward function is used: a coverage reward proportional to the discovered area, a success reward when a threshold value of coverage is achieved, and an overlap penalty for repeating coverage. The team members communicate extracted features from a CNN local feature extractor to one another.

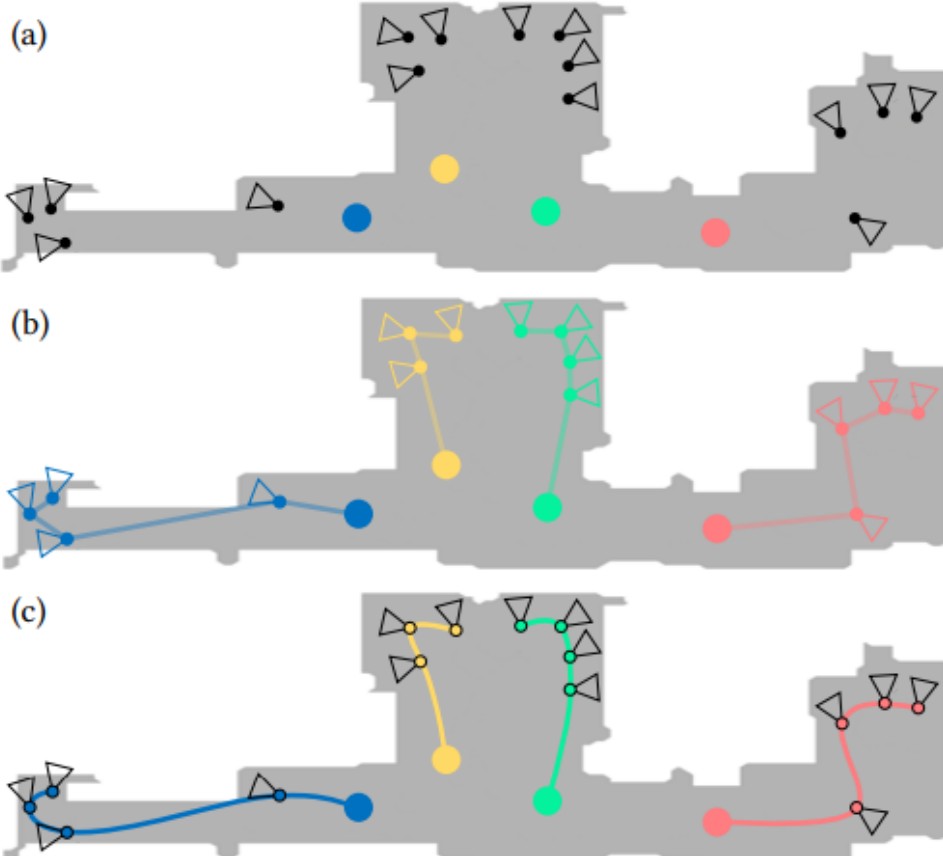

**Figure 15.** Paths over an iteration in the work of Dong et al. [54]. (**a**) The decomposed viewpoints and the robot starting positions. (**b**) The multiple TSP paths from the robots. (**c**) The smoothed paths for the robots.

*5.3. Discussion*

When discussing the applicability of decision making approaches to domain applications resembling OWT inspection, an initial question is whether model-based or model-free approaches are better suited to the task. In an OWT inspection, the orientation of the turbine is not tracked; multi-path error makes GPS positioning unreliable [38]; and in the case of floating OWTs, the entire structure can excur up to 35% of the depth of the mooring system [28]. A case could be made that a model-free approach would better account for the uncertainty of the turbines' poses. As previously proposed in Section 4.7, the OWT inspection task lends itself to 3D environmental models, with semantics labels for a search set representing the structure to be inspected.

Given a 3D voxel-based environmental model with a subset of voxels representing the search set, what planning approaches should be used to provide sensor coverage of the search set with a team of robots? The answer to this question depends on the assumed prior knowledge. Given the full environmental model a priori, a task decomposition approach such as that proposed by Dornhege et al. can be used [53]. However, Dornhege et al. were concerned with wheeled robots and the set of reachable voxels along the ground; applying the same approach to UAV OWT inspections would require a much larger set of reachable voxels, considerably increasing the computational cost of the set cover solution. One solution to reduce this problem is to apply bounds within which voxels are considered—not based on voxel reachability but on proximity to the search set. As for task allocation, the approaches reported in the literature are quite limited; approaches like greedy allocation or K-means clustering would work but may not provide near-optimal solutions. As for motion planning, given the assigned tasks for the team members, an open TSP approximation should be computed over the assigned task, generating a high-level path plan [53]. To follow this plan, a 3D cost map in the form of an ESDF should be used to prevent collisions with obstacles [55], and the path considers a tradeoff between the length and the cost. None of the task allocations take into account the dynamics of the OWT environment; for example, strong winds may increase the time to reach a given task, and this could be accounted for in terms of task cost. Additionally, none of the approaches considers the disruption of task performances itself, such as a strong gust of wind or a wave disrupting the image capture process for a given robot, which would require dynamic reassignment of the task to the team.

As previously discussed, an accurate model of the environment can be unrealistic in an OWT inspection due to the mobility of floating OWTs and the dynamic nature of the nacelle yaw and blade rotation. It may be necessary to treat the task as an exploration problem, following a similar approach to task decomposition as proposed by Dong et al. [54]. As with the approach of Dornhege et al., the issue with the approach of Dong et al. is its assumption of wheeled robots. Another issue is the approach's focus on exploring an entire environment rather than a search set of interests. This represents an area for future research. If the search set is not known a priori (as in the work of Dornhege et al. [53]), task decomposition requires the inference of the search set from sensor information (which requires the team to identify the OWT as the search set in an online manner). Assuming such an approach to identify the search set at each iteration, a frontier-based approach can be applied to generate tasks at the frontier of the known search set to identify more OWTs. Task allocation and motion planning can be achieved in much the same way as for model-based approaches. The predictable general geometry of the turbine could be utilised to allow these tasks to be generated. Such coverage of an unknown environment with significant prior knowledge, i.e., a structure-informed coverage of an unknown environment, is an interesting area of research that, as far as the authors are aware, has not received attention so far.

## 6. Coordination

This section aims to provide an answer to subquestion 3 from Table 2: *What is the most suitable strategy to effectively coordinate a multi-robot system for domain applications resem-*

*bling offshore wind turbine inspection?* Coordination has many definitions in the literature. Farinelli et al. [37] consider coordination to be cooperation according to which team members take actions in consideration of other team members "in such a way that the whole ends up being a coherent and high-performance operation". Yan et al. [36] defined it as multi-robot planning to deal with resource conflicts, be they conflicts in terms of space, tasks, or communication media. Cao et al. [77] defined coordination as follows: "Given some task specified by a designer, a the multiple-robot system displays cooperative behaviour if, due to some underlying mechanism (i.e., the "mechanism of cooperation"), there is an increase in the total utility of the system". While it is true that in all the approaches discussed, the team members coordinate to increase the utility of the system, this section focuses on coordination mechanisms necessary to resolve issues brought about by the dynamics of a task environment or the online nature of an approach. The decision making, even if aware of and, therefore, coordinating with other team members, is discussed in the previous section. Here, we discuss the necessary communication mechanisms required to facilitate this coordinated decision making. To succinctly evaluate the coordination mechanism in the reviewed works, they were charted as shown in Table 9, which shows whether the works take an online or offline approach to planning, as well as details of communication, whether the teams are heterogeneous or have an inter-team hierarchy, and whether fault tolerance is considered. These categories provide context concerning how the team members coordinate to complete their tasks.

**Table 9.** Table of data extracted for the coordination research question.

| Literature | Online/Offline | Communication | Hierarchy | Heterogeneity | Fault Tolerance |
|---|---|---|---|---|---|
| Ball et al. (2015) [40] | Online | Extrinsic | No | Homogeneous | Not discussed |
| Kalde et al. (2015) [45] | Online | Extrinsic | No | Homogeneous | Not discussed |
| Song et al. (2015) [46] | Online | Extrinsic | Dynamic hierarchy | Homogeneous | Not discussed |
| Colares and Chaimowicz (2016) [51] | Online | Extrinsic | No | Homogeneous | Not discussed |
| Dornhege et al. (2016) [53] | Offline | Extrinsic | No | Homogeneous | Not discussed |
| Perez-Imaz et al. (2016) [47] | Online | None | No | Homogeneous | Yes |
| Sharma et al. (2016) [48] | Online | None | No | Homogeneous | Not discussed |
| Masehian et al. (2017) [41] | Online | Extrinsic | Yes | Heterogeneous | Not discussed |
| Karapetyan et al. (2018) [42] | Offline | None | No | Homogeneous | Not discussed |
| Dong et al. (2019) [54] | Online | Extrinsic | No | Homogeneous | Not discussed |
| Zhang et al. (2019) [49] | Online | Extrinsic | No | Heterogeneous | Not discussed |
| Bramblett et al. (2022) [52] | Online | Extrinsic | No | Homogeneous | Not discussed |
| Kim et al. (2022) [44] | Online | Extrinsic | No | Heterogeneous | Yes |

**Table 9.** *Cont.*

| Literature | Online/Offline | Communication | Hierarchy | Heterogeneity | Fault Tolerance |
|---|---|---|---|---|---|
| Tang et al. (2022) [43] | Online | Extrinsic | No | Heterogeneous | Not discussed |
| Bartolomei et al. (2023) [55] | Online | Extrinsic | No | Homogeneous | Not discussed |
| Yu et al. (2023) [50] | Online | Extrinsic | No | Homogeneous | Yes |

*6.1. Collaboration in Decentralised Planning*

In centralised approaches, a central planner assumes knowledge of the state of the environment and dictates tasks based on this global view. Such an approach is powerful and can find optimal solutions if feasible, but this is very rarely feasible. Communication issues or failure of the planner have a catastrophic effect on online coverage with a centralised planner. Hence, many approaches attempting to perform coverage with communication restraints implement distributed approaches to the problem. Additionally, decentralised approaches can handle a larger number of robots by distributing the computation among robots. In our review, eight works were identified using decentralised planning. As discussed earlier, Kalde et al. [45] considered a decentralised frontier-based approach. The robots share a map and their locations. With the locations of the robots shared, a task robot cost matrix is formed by each robot using the map; however, the matrix only considers robots local to the computing robot to optimise the assignment. While centralised task allocation was considered by Song et al. [46], a decentralised approach was adopted to handle unequal task sizes. In this approach, once a robot completes its initially assigned task, it starts a cooperative game with robots local to it in need of help. Cooperative games are one of the state-based potential games described by Marden [78]. In their case, the cooperative game is said to be made up of coalitions of two robots, each with a payment balancing the distance to the task of the receiving robot and the remaining uncovered cells in the task. Given this, the optimal coalition is selected by the initiating robot, which assists in the completion of the receiving robot's task. This process is shown in Figure 16.

In Colares and Chaimowicz's work, a single utility function taking into account robot positions was used to coordinate decision making [51]. The robots communicate implicitly through a camera identifying their positions and poses in the locale. After this implicit identification, the robots share their maps and pose information, and the initial robot communicates an estimated pose for the spotted robot relative to itself. Using this information, map stitching is used to combine the map information for both robots. The cost function used to assign tasks for a given robot is implemented in a decentralised manner, with a cost function composed of the value of a task based on its neighbours; the distance to the task; and—most relevant to this section—a coordination factor disincentivizing allocation of tasks close to the identified neighbouring robots. Bramblett et al. [52] considered exploration and task coverage in an unknown environment under the constraint of a limited communication range. Therefore, the team is required to rendezvous intermittently to share environmental and task information. For the exploration phase, K-means clustering is used to assign task areas to robots whenever they can communicate. The clusters are auctioned in a centralised manner. A rendezvous mechanism is used whenever all robots are connected, which finds a valid navigable point with minimal distance to the centroids of the robots' partitions. After some time, the robots navigate back to this rendezvous point to share information. If a task is discovered during exploration, a rendezvous policy representing the cost of rendezvousing is formed from the potential path to the rendezvous and the unknown space it passes through balanced with the subtraction of the global expected path length from the length of the explored path and the task length. The approach used by Tang et al. [43] for coordination takes the form of the use of two classes of robot for the coverage problem. The authors adopted worker station approach to coverage where the workers have limited energy, while the stations have unlimited energy and the ability

to replenish the workers. They considered this problem as a multi-agent reinforcement learning problem. The robots can communicate and use this communication to form their observations of the environment. The observation space is composed of three components: zero-range observations are the position velocity and energy of the agent; perception-range observations provide information about obstacles and agents within the perception range; and communication-range observations include information about the agents within the communication range. The authors made use of centralised training–decentralised execution (CTDE), in which their critic network has full knowledge of the environmental state but the actions of an individual agent are based on local observations. Additionally, a two-stage curriculum is used for training, with a simple environment of one actor and one station used initially until convergence, followed by an environment with two stations and four workers [79]. Multi-layer perception policy networks are shared between robots of the same class but differ between the worker and the stations to account for their differing abilities. A visualisation of their deep reinforcement learning pipeline is shown in Figure 17.

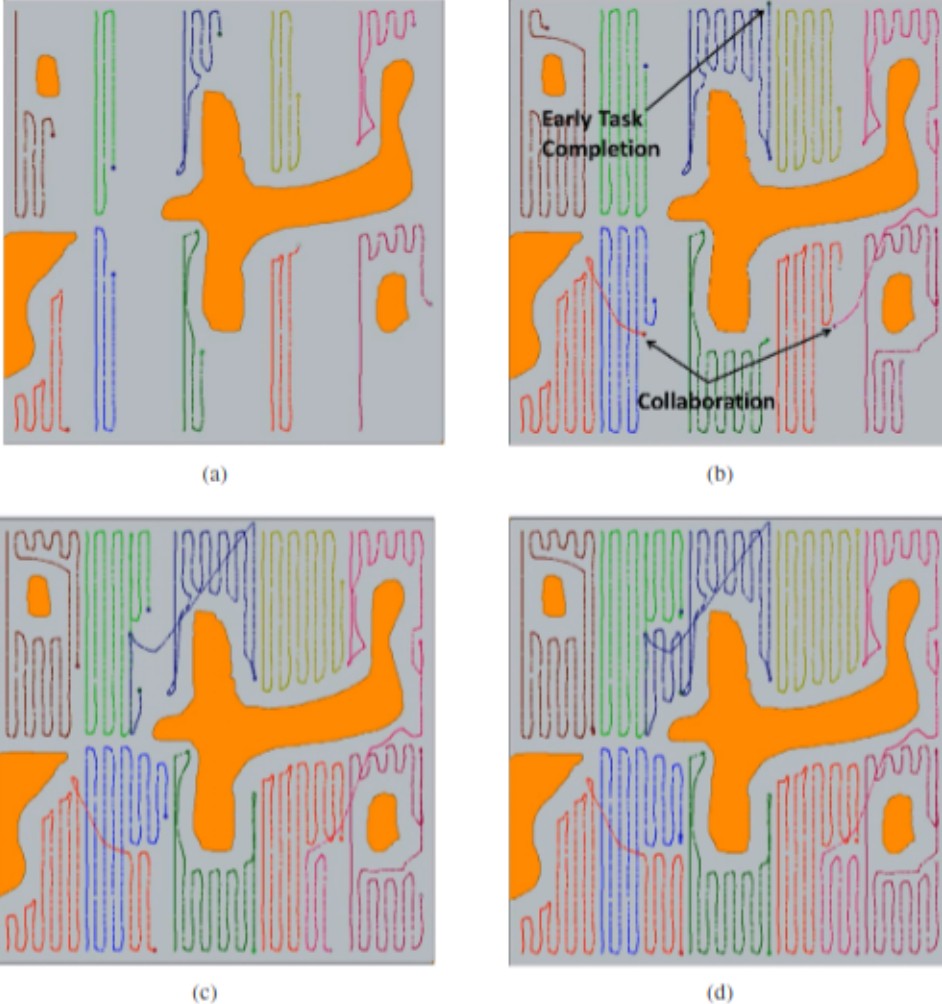

**Figure 16.** Kalde et al.'s cooperative collaboration game: (**a**) The robots start coverage (**b**) Several robots finish coverage and move to assist others (**c**) The areas are split between the collaborating robots (**d**) Coverage is completed [45].

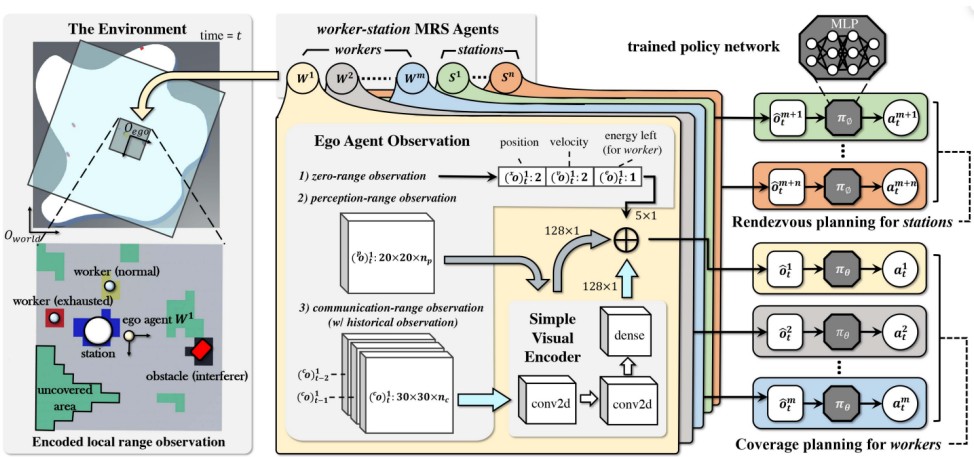

**Figure 17.** Tang et al.'s deep reinforcement learning pipeline [43].

In the work of Bartolomei et al. [55], a team of robots completed exploration with roles, exploration, and collection. The members of the team can vary roles based on the needs of the team. Exploration involves seeking large patches of unexplored frontier, while collection prioritises small, unexplored sections of the map surrounded by covered areas. The robots, by standard, take on the role of an explorer, but given a threshold number of disjoint unexplored regions close to the robot, switch to collector mode. Another approach that considers the problem of multi-robot reinforcement learning for exploration was considered by Yu et al. [50]). The authors noticed that the existing literature primarily focused on agents acting in a fully synchronous manner, which is a problematic assumption for real-world adoption. As such, they adopted an asynchronous multi-agent proximal policy optimisation approach to training. Each robot has its policy network; therefore, behaviour varies between members. To better facilitate communication between the robots, a CNN was used for feature extraction from the local environmental map, and these features were then shared between members of the team. To further facilitate collaboration, the reward function takes into account the overlap between the coverage of the robot and the rest of the team, discouraging repeated coverage of the same area.

### 6.2. Communication

Nordin et al. [16] identified several issues with communication in offshore wind turbine environments. For example, there is likely to be no cellular network coverage due to the distance from land. Furthermore, normal satellite communication has a high latency that would hinder online planning, and although there now exists real-time satellite communication in the form of the Inmarsat SwiftBroadband satellite service, it may be hindered due to proximity to the towers [80]. The authors' proposal is the use of USVs to connect to a satellite service positioned away from the towers, enabling communication with the UAVs through an ad hoc Wi-Fi network. Communication was classified into two categories by Matric [81]. Direct communication is purely communicative, transmitting data from one agent to another or to a central planner. Indirect communication is based on observation; a robot could, for example, sense the tracks of another robot, communicating the fact that an area has been visited. While all online cooperative approaches considered in this review make use of explicit communication, some additionally make use of implicit communication. In the approach of Ball et al. [40] for broadacre agriculture, the real-world implementation uses a 3G mobile data connection to the Internet for communication between the robots and a central planner using ROS middleware. A map is shared between the robots in the approach proposed by Kalde et al., although the communication mechanism was not described [45]. Song et al. [46] made use of a player/stage simulator, which allows modules to communicate through TCP. Colares and Chaimowicz [51] used ROS as middleware for their real-world experiments. Communication was not discussed by Dornhege et al. [53]. The approach of Perez-imaz et al. [47] involves robots communicating

their position with a central planner as an approach to fault tolerance; this communication is achieved again through ROS. Masehian et al. [41] considered communication between the robots and the central planner to be of unlimited bandwidth, assuming ideal conditions. Karapetyan et al. [42] assumed no communication capabilities, with a purely offline approach. Dong et al. [54] considered communication between robots and a central planner. Zhang et al. [49] also considered a centralised planner, although details of the implementation are sparse. Bramblett et al. [52] made use of a disk constraint to simulate communication-range constraints. Kim et al. [44] made use of a robofleet for communication [82], with communication used for fault detection among the team. Tang et al. [43] also considered the communication range.

### 6.3. Fault Tolerance

Fault tolerance is a crucial aspect of building robust multi-robot systems. Multi-robot systems provide inherent redundancy by allowing other robots to complete the tasks previously assigned to faulty robots. A reality of working outside of simulation is that eventually, failure occurs. Robot failure was explicitly discussed in two of the reviewed works: that of Perez-imaz et al. [47] and that of Kim et al. [44]. Other approaches, such as frontier-based exploration approaches, might offer inherent robustness to failure as a result of iterative planning. In Perez-imaz et al.'s work [47], when a robot failure occurred, hexagon cells were reallocated to members of the team. Similarly, in the work of Kim et al., when a robot failure was detected, the system recomputed the coverage task decomposition with a smaller team size [44]. However, this still results in repeated coverage of areas already covered by the failed robot. The reallocation mechanism used by Kim et al. is shown in Figure 18. Both these works focus on recomputing offline task allocations.

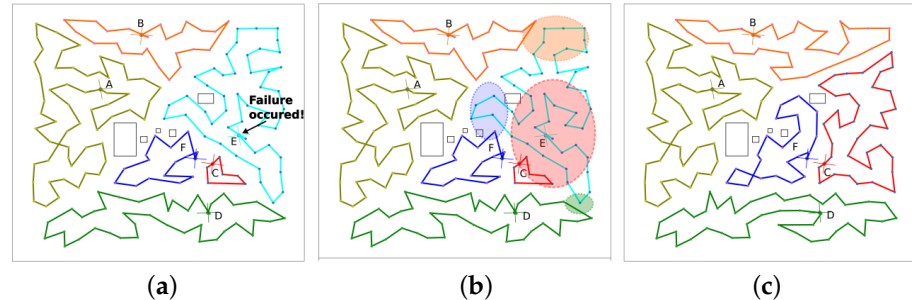

**Figure 18.** Kim et al.'s fault tolerance reallocation [44]. (**a**) The robot with the cyan path experiences failure. (**b**) The waypoints are reassigned to neighbouring robots. (**c**) The TSP paths for the new allocations are computed, ensuring coverage.

### 6.4. Discussion

It is difficult to imagine a robust real-world coordination framework using only the approaches discussed in this work. Such a coordination framework would need to account for communication downtime, robot failures, and possibly heterogeneous capabilities. One noticeable trend, albeit with a small sample size, is a focus on reinforcement learning approaches in recent years. Yu et al. [50] note that reinforcement learning approaches, when compared to traditional planning approaches, can effectively produce complex strategies and, after training, prove computationally inexpensive. Regardless, the majority of the literature considers planning-based approaches. There seems to be potential for future research on both classes of algorithms. Fault tolerance and dealing with communication constraints are open avenues for research, with only two of the reviewed works explicitly considering faults [44,50]. Coordination for heterogeneity has received very little focus from the research community, with current works concentrating on worker–station relationships [43,49] or sensing range and locomotion speed [44]. Masehian provided a highly specific case of robot mapping with different forms of sensors [41], but beyond that, there has been no work focusing on robots' semantic capability regarding completing tasks or traversing the

environment. Coverage of tasks with semantic requirements for both completion and area traversal by heterogeneous teams is still an open area of research. Dynamic environments were rarely the focus of the reviewed works, with only Kalde et al. [45] considering such, with mobile obstacles. Dynamic environments are a potentially interesting area of research for OWT inspection due to the mobile nature of wind turbines, even more so for floating offshore wind turbines. Another area of interest that has not been considered is dynamic tasks. Considering the problem of covering a single OWT, the task of visually covering the blades may not be localised at one static coordinate. If the wind turbine is in operation, the coverage task will be constantly and predictably moving. Such a dynamic coverage task is another potential direction for future research.

## 7. Future Work

None of the works reviewed in this paper can enable coverage for OWT inspections alone. A comprehensive multi-robot coverage system would require the combination and extension of existing techniques. Several potentially useful aspects of the reviewed approaches have been identified in the previous sections. In this section, we attempt to synthesise the identified approaches and their limitations with respect to several areas of future OWT inspection research.

### 7.1. Task Decomposition with Areas of Interest

One aspect of OWT inspection coverage that has not been addressed in existing research is coverage with varying required degrees of quality. In OWT inspection, certain sections of the turbine may require greater focus than other sections. Usually, the tower is of less interest than the turbine blades. To address this, it is necessary to select and extend an existing environmental representation to account for varying coverage requirements across the structure. One method for achieving this would be through a bespoke semantic label applied to sections of the environment. Assuming a voxel-based representation, this may be a property for each voxel that specifies, for example, a required proximity for coverage. This semantic label would then need to be accounted for when decomposing the task into a set of views, only considering a voxel covered if a view fulfils the requirement encapsulated within it. An alternative approach is the use of multiple resolutions depending on the degree of interest in a section. This would not inherently apply proximity requirements, but it would ensure more thorough coverage within a specified region. This could be achieved through an Octomap [62] and would allow for the use of the approach proposed by Dornhege et al. [53] without modification. Combining these two approaches may prove even more efficient in ensuring both thorough and high-quality sensor coverage. However, this all assumes the area of interest is known a priori. To identify areas of interest in an unknown environment, some form of semantic area detection would be necessary, maybe through object detection techniques.

### 7.2. Limited Knowledge Approach

An interesting area of research is the possibility of using the known geometry of the turbines in an otherwise unknown environment. The geometry of a turbine is always available before an inspection, and intuitively, an approach should be able to exploit this knowledge. None of the reviewed works considered the case of geometric structural knowledge in an otherwise unknown environment. The most obvious use case for this is in floating OWT inspection, where the turbines have drifted from the centre of the moorings; however, just because the turbine has moved a certain amount does not mean that the environment is now completely uncertain. This could be achieved by considering a problem of two layers: exploration within a small subarea of the environment to localise the turbine followed by model-based coverage of the now-known structure.

### 7.3. Dynamic Tasks

In all the reviewed works, the area or structure to be covered was static. Given a moving structure such as OWT blades, task planning and motion planning would be significantly complicated, and a novel environmental representation would be necessary to represent the moving tasks. One possible solution for the blades is to use one team member to constantly observe and track the blades' positions, then use other team members to complete the coverage of the required proximity and quality. This problem identifies one key issue with using a voxel-based representation alone, in that voxels tend not to represent semantic objects but just occupancy, so when the physical object moves, some mechanism would be necessary to ensure that any label is transferred to the new voxel representing that physical object.

### 7.4. Limited Communication

As discussed by Nordin et al. [16], communication is an issue in the OWT environment. While most of this work has been focused on UAV coverage of turbines, it is the case that UAV batteries are currently limited, and any feasible implementation would require the use of USVs for UAV deployment. As Nordin et al. suggested, the use of a USV may also play a role in solving the communication problem for OWT coverage. This slightly resembles the worker–station approach of Tang et al. [43]. An approach that strategically places a USV distant enough from the turbines for satellite communication interference from the turbines while providing a temporary wireless network for the UAVs in the team may solve this problem. This would require a new approach to planning, accounting for USV placement and possibly requiring a rendevous mechanism with UAVs working outside of the network, then returning.

### 7.5. Heterogeneous Sensing/Locomotion Capabilities

Heterogeneity among robots was lightly touched upon in the reviewed work, but to fully harness the capabilities of a diverse team, new planning approaches would be necessary. Sensing heterogeneity can be implemented in the sense of team members with different sensor specifications, such as some members with cameras specialised for closeup photography or carrying thermal cameras. Alternatively, there is homogeneity in locomotion capabilities, as some robots may fly, like UAVs, and some cannot and are limited to the surface, such as USVs. If tasks are going to be shared between these members, the capabilities should be taken into account. The capabilities of the team members should be considered through all aspects of the OWT coverage problem. The environmental representations should encapsulate the requirements of both tasks and the traversal between them. Task decomposition should derive the requirements for a task from the information at hand. Tasks should only be allocated to robots able to complete them, and the capabilities of the robots should be accounted for when grouping tasks. Finally, motion planning should plan paths and trajectories taking into account the capabilities and the kinematics of the robot being planned for.

### 7.6. UAV Structural Coverage

None of the approaches reviewed herein consider 3D sensor coverage with UAVs, which is a necessity for the OWT coverage task. The approaches that do consider 3D structural inspection assume a prior model and use offline planning. None of the approaches consider heterogeneous tasks or locomotion capabilities, which would be essential for heterogeneous structural coverage. If blade coverage is to be performed while the OWT is in use, it would be necessary to represent the blade as a moving task and track the blade's position; however, none of the studies reviewed herein considered this issue. To tackle the problem of 3D sensor coverage with UAVs assuming prior knowledge of the environment, one may use the ray-tracing voxel-based task generation as in the work of Dornhege et al. [53]. Rather than the reachable voxels being located along the ground, it would be necessary to ensure that the sensor is a certain distance from the turbine

surface; without this, the set of reachable voxels would be very large and, therefore, inefficient for computation. This same approach could be extended to perform in an online exploratory manner; however, 3D structural exploration was not considered in any of the reviewed works.

## 8. Conclusions

In this work, a review of the literature on multi-robot coverage concerning OWT inspection was carried out. The PRISMA 2018 Scoping review methodology was followed to standardise the review process, along with the PICo framework to form and model the research questions. These approaches for standardizing the review process are rarely used in computer science and even less so in robotics literature. However, such systematic processes are essential for providing a scientific review that the reader can repeat themselves and obtain the same or representative data. The retrieved works were then systematically analysed with respect to the formed research questions and discussed. This work applies not only to OWT inspection scenarios but also to scenarios resembling offshore wind inspection. It is important to note that coverage planning algorithms are far from the only hurdle in putting autonomous offshore inspections into practice; coverage path planning structural inspection should be considered one component of a larger system. As of the time of writing, drone battery durations would not be sufficient to enable their use alone from shore. To enable the long-term autonomy required for wind farm inspections, an approach to charging drones in the field would be necessary, such as that proposed by Han et al. [83], in which drones are launched from a USV with the capability of charging the drones when necessary. Several areas for future research were suggested herein. Decomposing the coverage task concerning areas of particular interest would facilitate more detailed coverage, allowing for focus on areas of the turbine most prone to failure or where failure is most critical. The use of existing knowledge of the turbine geometry without further knowledge of placement or pose is particularly applicable to floating OWTs. Dynamic tasks, where tasks might move within the environment, and the importance of keeping track of covered and uncovered moving structures were also discussed. We also addressed the limitations of communication around large OWT structures that may affect satellite communication. Furthermore, we considered heterogeneous capabilities in the team, in terms of both sensing and locomotion, facilitating complex planning for teams aware of capabilities. Finally, we extended existing surface robot voxel-based approaches to UAVs while minimising the computational complexity due to the large size of the accessible space.

**Author Contributions:** Conceptualization, A.J.I.F., M.G., A.A., H.S. and S.S.; methodology, A.J.I.F., M.G., A.A., H.S. and S.S.; formal analysis, A.J.I.F.; investigation, A.J.I.F.; data curation, A.J.I.F.; writing—original draft preparation, A.J.I.F.; writing—review and editing, A.J.I.F.; supervision, A.J.I.F., M.G., A.A., H.S. and S.S.; project administration, A.A. All authors have read and agreed to the published version of the manuscript.

**Funding:** This research is assisted by the EPSRC DTP (ORE) at the University of Plymouth.

**Data Availability Statement:** Please find the data for this work at the following link: https://osf.io/yq5rg/?view_only=9d771c0b874d40aa8bbc6011752bd8e0 (accessed on 13 November 2023).

**Conflicts of Interest:** The authors declare no conflicts of interest.

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
