# Peer review of "Multi-Robot Coverage Path Planning for the Inspection of Offshore Wind Farms: A Review"

_drones, doi:10.3390/drones8010010_

Round 1

Reviewer 1 Report

Comments and Suggestions for Authors

The paper investigates the multi-robot coverage path planning problem to address the inspection of offshore wind farms. The related works are analyzed through three aspects of coverage approaches. The paper carefully analyzes the methods for the robot coverage problem in this specific domain. Some solutions can be used to address the coverage problem of UAVs, while some of them are mainly used to resolve the coverage problem of mobile robots. In general, the paper is well organized and easy to follow.

One issue that need to be further addressed is about the MULTI-ROLBOT solutions for the coverage problem. For a robot team, the solutions can be centralized or decentralized. Normally, the task allocation approaches in the robot domain rarely consider the interference in ocean environments. For example, an unmanned ship may be disturbed by environmental uncertainties during executing an allocated task. In this case, for the coverage problem, the dynamic changes during task execution need to be considered in real-time, not just a static planning problem.

Reviewer 2 Report

Comments and Suggestions for Authors

Wind power is getting mainstream, and maintenance is of high interest due to extreme weather condition in offshore environment. So, the topic of the paper is relevant. However, there is a contrast between the actual application (OTM) and discussion, which, despite all the effort, is quite generic. The authors, clearly, did a significant amount of work, unfortunately, the value for OTM is a not obvious; at best limited if any.

Some general comments:

·         Objectives of OTM inspection are not clearly spelled out. Is it to LiDAR scan and camera image the entire surface of the structure? Or are there areas of higher interest? What is the required performance of the sensing platform? Etc. Answering these questions can define the place where a robot or multi-robot solution can be optimized.

·         Why not addressing inland wind turbine case? At least, to provide a short discussion on it, as clearly the OTM is a generalization of this case to a more challenging environment. Any work done there? Some elaboration on it would be useful or rather needed.

·         Why it is not essential, but some connection to the feasibility of robot inspection would be good to have. How far are from implementations? UAVs mentioned a few times, so is it realistic to send a swarm of UAVs to an offshore wind farm? Given the short flight times of the UAVs, it doesn’t sound even close to reality. Of course, you may still write a paper, but then be careful about the scope of it and avoid creating a false impression that this is the only thing missing from an automated OTM.

·         Since wind turbines have very typical geometry, how these specifics can be considered and/or exploited for OTM? This would be not only of interest but required to arrive to an optimal solution.

·         Any collaborative navigation requires good communication, which is a challenge even inland but definitely offshore. There is some discussion on that topic but is on highly theoretical level with no connection reality. Again, this ties to the feasibility UAV-based surveying of OTMs.

·         Finally, the OTM may sound as an exotic topic which is interesting yet a covert title, as the discussion is more of generic nature; yes, somewhat filtered to the criteria which are very loosely connected to the specifics of OTM.

Some specific comments (not a complete list):

·         Lines 127-129 has a statement which gently challenges the objective of the paper. Yes, Lines from 131 provide some explanation, but appears to be weak and somewhat artificial.

·         Only past 2015 literature is covered. Why robotics is heavily dependent on technology, yet, for example, path finding/planning algorithms have not changed much recently, so a broader timespan with more specific filtering may be considered.

·         Lines 194-196 says “To perform inspections between turbines, the team can’t rely on LIDAR sensor data alone for large portions of the navigation, the environment is too sparse and the distances between features are too great.” What about GPS/GNSS? Isn’t it the first line of navigation? Have the authors seen a UAV without GPS? While LiDAR can be used for local or rather relative navigation, it is hardly ever considered as a navigation sensor.

·         Table 6, the font is just too tiny, and can’t be read at normal scale. BTW, there are a few other tables with the same problem.

·         Fig 8 shows the clearance for the blades, but what about blade length and tower height or even width parameters? The figure is not providing more what even laymen know about these structures.

·         5.1.2 is long and dry text, which is absolutely not structured. There are four pages with three figures of which two are not conveying much information. For example, it would be good to see robot trajectories and other informative and attractive visuals. The text is a sequential discussion of topics which offers not much to the reader as it lacks the synthesis, the joint evaluation of the various methods. This comment applies to other sections too.

·         Fig 14 would be fine, except the extent how a lawnmower example can be applied to OTM is quite questionable. One is a 2D problem in a closed space, while the other is 3D where the objects of interest are sparsely distributed, clearly, requiring a more complicated path.

·         Fig 15 the smoothed trajectories seem to be slightly different, as the yellow and green ones appear less smoothed. Of course, it is not a critique rather just an observation.

·         In the first paragraph of 5.2, Why is it even a problem to locate the OWTs? GPS was mentioned earlier. So, there is no need to chase the floating ones and/or static ones. BTW, they can even broadcast their location. Path planning doesn’t appear to be an issue for OTM.

·         The conclusion is light on OTM and written on a fairly high level and provides little new information for a professional who is the robotic field.

·         The English is good, so only a routine check is needed. For example, Line 622 “asks” should be “tasks”, or “K-means” is incorrectly “K-mean” at some places.

Reviewer 3 Report

Comments and Suggestions for Authors

The paper describes offshore wind turbine inspection research as a multi-robot coverage path planning problem. Furthermore, according to the environmental modelling decision-making, and coordination of coverage path planning problems, the paper summarizes existing research work. In addition, this paper analyzes the shortcomings of existing methods in offshore wind turbine inspection and proposes corresponding future research. However, this paper has many problems. The specific problems are as follows:

1. The introduction of this paper fails to explain the coverage planning problem for offshore wind turbine detection clearly. In addition, how is the coverage planning problem in this scenario different from the traditional coverage planning problem? The author should combine a schematic diagram to explain the coverage planning issues for offshore wind turbine detection.

2. Section 3 of this paper provides an overview of existing research. But its expression lacks organization, which is difficult to understand. Furthermore, the environmental modeling, decision-making and coordination include all research work on coverage path planning problem? The author should illustrate the current research status according to a schematic diagram.

3. The Section 5 of this paper includes model-based methods, non model-based methods, and path planning methods. What is the difference between the above content? In addition, Section 5.1.1 introduces the classification methods of other scholars, which seems to have little effect. The author should delete irrelevant content to highlight the core part.

4. The description of future research in this paper is too confusing, which cannot clearly demonstrate the motivation for future research. The author should rewrite future research.

Comments on the Quality of English Language

The paper describes offshore wind turbine inspection research as a multi-robot coverage path planning problem. Furthermore, according to the environmental modelling decision-making, and coordination of coverage path planning problems, the paper summarizes existing research work. In addition, this paper analyzes the shortcomings of existing methods in offshore wind turbine inspection and proposes corresponding future research. However, this paper has many problems. The specific problems are as follows:

1. The introduction of this paper fails to explain the coverage planning problem for offshore wind turbine detection clearly. In addition, how is the coverage planning problem in this scenario different from the traditional coverage planning problem? The author should combine a schematic diagram to explain the coverage planning issues for offshore wind turbine detection.

2. Section 3 of this paper provides an overview of existing research. But its expression lacks organization, which is difficult to understand. Furthermore, the environmental modeling, decision-making and coordination include all research work on coverage path planning problem? The author should illustrate the current research status according to a schematic diagram.

3. The Section 5 of this paper includes model-based methods, non model-based methods, and path planning methods. What is the difference between the above content? In addition, Section 5.1.1 introduces the classification methods of other scholars, which seems to have little effect. The author should delete irrelevant content to highlight the core part.

4. The description of future research in this paper is too confusing, which cannot clearly demonstrate the motivation for future research. The author should rewrite future research.

Reviewer 4 Report

Comments and Suggestions for Authors

This paper provides a scoping review for multi-robot coverage from perspective of decision-making, environmental modeling, and coordination. Especially, the discussions about application in OWT are carried out for each scoping review. This work is useful for the research of multi-robot systems. However, the paper should be checked thoroughly. Some minor revisions should be made. E.g. In table 3, the URL of “Web of Science” should be revised.

Round 2

Reviewer 2 Report

Comments and Suggestions for Authors

The authers have done a good job to improve the manuscript, which is now ready for publication.

Lines 878-880, obviously, should be deleted.

Reviewer 3 Report

Comments and Suggestions for Authors

Modified as required.

Comments on the Quality of English Language

Minor editing of English language required.